# Sub-angstrom strain in high-entropy intermetallic boosts the oxygen reduction reaction in fuel cell cathodes

Xueru Zhao[1,9], Hao Cheng [2,9], Lijun Wu [3], Qi Zhang[4], Xiaobo Chen [5], Nebojsa Marinkovic [6], Chenzhao Li [4], Sha Tan [1], Enyuan Hu [1], Lu Ma [7], Yimei Zhu [3], Jian Xie [4,8] & Kotaro Sasaki [1] ✉

The strain effect of high-entropy intermetallic (HEI) catalysts on oxygen reduction reaction (ORR) performance remains largely unexplored, primarily due to the significant challenges associated with characterizing and calculating the intricate local coordination environments. Here, we design a nitrogen (N)-doped $L1_0$-ordered PtCoNiFeCu intermetallic catalyst supported on Ketjen-black carbon (N-HEI/KB), and reveal the origin of the sub-angstrom strain in N-HEI and its impact on ORR performance by combining atomic-scale characterization and theoretical calculations. The synergistic interplay of the sub-angstrom strain, the pinning effect of metal-N bonds, and the high-entropy effect contribute to the competitive stability of N-HEI/KB catalysts, providing high current density of 1388 mA cm$^{-2}$ at 0.7 V after 90,000 cycles even under harsh heavy-duty vehicle conditions. These findings broaden the avenues for designing high-performance high-entropy intermetallic cathode electrocatalysts.

With the development of electric vehicles, the polymer electrolyte membrane fuel cells (PEMFCs) have been playing an increasingly crucial role as the power sources in renewable energy applications[1,2]. Several Pt-based alloy catalysts, such as PtCo[3] and PtNi[4] catalysts, have been identified as promising candidates to meet the membrane electrode assembly (MEA) targets established by the U.S. Department of Energy (DOE) for passenger light-duty vehicles. However, as the attention shifted towards more demanding heavy-duty vehicles (HDVs), new challenges arose regarding the durability and efficiency of the oxygen reduction reaction (ORR) catalysts[5–8]. Addressing the long-term stability of Pt-based alloys hinges on mitigating Pt degradation and reducing the dissolution/oxidization of 3$d$ transition metals (TMs) during the extremely long fuel cell operation in HDVs[9,10].

Pt-based intermetallic compounds have received increasing attention on ORR catalyst research for the following two reasons: (i) Strong 3$d$–5$d$ orbital coupling between TMs and Pt exhibits superior resistance to TM leaching, leading to improved stability[11–13]. (ii) The formation of ordered intermetallic structures induces large lattice strain that results in changes in bond lengths and coordination environments; this, in turn, affects the electronic state of Pt and boosts the catalytic activity[13–15]. Compared to binary intermetallic Pt-based alloys, multiple-component high-entropy intermetallic (HEI) catalysts (that contain five or more elements) possess improved thermodynamic and chemical stability due to the high configurational entropy and sluggish atomic diffusion[16,17], making them ideal candidates for ORR electrocatalysts[18–20]. So far, a comprehensive understanding of the relationship between structure and electrocatalytic performance of

[1]Chemistry Division, Brookhaven National Laboratory, Upton, NY, USA. [2]Department of Applied Physics, The Hong Kong Polytechnic University, Hung Hom, Kowloon, Hong Kong, China. [3]Condensed Matter Physics and Materials Science Division, Brookhaven National Laboratory, Upton, NY, USA. [4]School of Mechanical Engineering, Purdue University, West Lafayette, IN, USA. [5]Department of Mechanical Engineering & Materials Science and Engineering Program, State University of New York at Binghamton, Binghamton, NY, USA. [6]Department of Chemical Engineering, Columbia University, New York, NY, USA. [7]National Synchrotron Light Source II, Brookhaven National Laboratory, Upton, NY, USA. [8]School of Materials Engineering, Purdue University, West Lafayette, IN, USA. [9]These authors contributed equally: Xueru Zhao, Hao Cheng. ✉e-mail: ksasaki@bnl.gov

HEI catalysts remains elusive. The design of advanced HEI catalysts is critical to elucidate their catalytic mechanisms.

Although the distribution of constituent elements in a high-entropy alloy (HEA) was thought to be random, recent studies have reported that local atomic segregation and cluster formation are observed in HEAs[17,21–23]. The intermetallic structure observed in ternary $L1_0$–PtCoNi is expected to produce anisotropic strain on surface Pt atoms, which can significantly improve the ORR performance of the catalyst[24]. However, previous research on Pt-based intermetallic catalysts has primarily focused on the strain effects on Pt atoms, with little attention given to the strain interactions between different TMs within the TM layers. This oversight highlights a gap in understanding the full potential of these materials. We hypothesize that HEI catalysts have the combined properties of HEAs and intermetallic compounds. In HEA catalysts, traditional lattice distortions are caused by mismatches of ionic size, mass and valence-electron configuration. The enhancement in catalytic performance of HEA primarily originates from entropy-driven structural stabilization and synergistic interactions among the constituent metallic elements[25,26]. The enhanced mixed configuration entropy of HEA contributes to the high-entropy effect, thereby enhancing the thermodynamic stability of catalysts[27–29]. On the other hand, in HEI catalysts, in addition to the strain that arises from atomic size differences, there exists the displacement of TM atoms in the TM layers, known as sub-angstrom strain. This occurs because the TM atoms in HEI catalysts occupy the sites in each sublattice with local clustering/segregation, driven by the stabilizing effect of Pt layers. This local arrangement potentially influences the electronic structure of HEIs and introduces additional sub-angstrom strain, consequently achieving improved ORR activity and durability. Moreover, the sub-angstrom strain vectors in HEI catalysts introduce anisotropic strain, distinguishing them from bimetallic PtM (M = Co, Ni, Fe, Cu, etc.) intermetallic compounds.

In this work, we synthesize nitrogen (N)-doped PtCoNiFeCu intermetallic nanoparticles (NPs) supported on Ketjenblack (KB) carbon (N-HEI/KB) via a facile thermal annealing process. The atomic-resolution images reveal atomic displacements of TMs in the $L1_0$-ordered structure of HEI due to local atomic segregation, and we suggest that N dopants further amplify these displacements through interstitial doping effects. Combining in situ X-ray absorption spectroscopy (XAS), density functional theory (DFT) calculations, and universal machine learning interatomic potential (uMLIPs), we elucidate the origin of the sub-angstrom strain in N-HEI and its consequential effect on catalyst stability. This highly stable structure of the N-HEI/KB catalyst exhibits superior long-term durability as characterized by both rotating disk electrode (RDE) and MEA. In particular, the current density of N-HEI/KB at 0.7 V reaches 1388 mA cm$^{-2}$ after 90k accelerated durability test (ADT) cycles, surpassing the latest DOE target for HDV conditions (1300 mA cm$^{-2}$ at 0.7 V after 90k cycles) as specified by the Million Mile Fuel Cell Truck (M2FCT) consortium[30].

## Results

### Structure and morphology characterization

In situ synchrotron X-ray diffraction (XRD) measurements were conducted to monitor the structural evolution of HEI ($L1_0$-PtCoNiFeCu intermetallic) NPs from HEA during annealing under 10 vol% $H_2$/He. We observed three temperature regions with the gradual formation of the HEI NPs. Pt NPs were reduced in the stage I (temperature region below 220 °C); HEA was generated in the stage II (220–740 °C); and in the stage III (above 740 °C), HEI was generated. As shown in Fig. 1a, b, reduction of the metal precursor (mainly Pt) occurs at around 155 °C, evident from the (111) peak at 15.86° in Fig. 1b. The Rietveld refinement analysis shows that the lattice constant of the sample at 155 °C is around 3.891 Å (Supplementary Fig. 1a), which closely matches the value of pure Pt metal (3.920 Å),

but is much larger than those of Co (3.537 Å), Ni (3.524 Å), Fe (2.866 Å), and Cu (3.615 Å). This value shows negligible change until 220 °C, pointing to the formation of $Pt^0$-rich phases in the NPs during stage I. The $Pt^0$-rich NPs act as seeds for the formation of high-entropy materials, and the observation is in accord with our previous study[31]. At around 220 °C, a new (111) peak appears at 16.81° (Fig. 1b); the peak corresponds to a lattice constant of about 3.673 Å, which is significantly smaller than that of Pt, suggesting the alloying of 3$d$ TMs with the Pt-rich NPs. As the annealing temperature rises, the intensity of Pt peaks gradually declines and eventually disappears (black dotted arrow in Fig. 1b), while the peaks representing HEA increase concurrently (black solid arrow in Fig. 1b). Accordingly, the lattice constant of the Pt-rich phase decreases with rising temperatures at stage II, and that of the alloy phase increases (Supplementary Fig. 1a). This suggests a progressive increase in the level of alloying as the continuous diffusion of TM atoms into the face-centered cubic (fcc) lattice of Pt and a gradual compositional homogenization[31]. Furthermore, a characteristic weak peak at around 13.1° emerged at 740 °C. This feature corresponds to the (110) superlattice peak (blue arrow in Fig. 1b), indicating the structural transition from a disordered PtCoNiFeCu alloy (fcc) to an ordered $L1_0$-PtCoNiFeCu intermetallic structure (Fig. 1c). During stage III, two distinct phases were identified: a HEA phase with relatively larger and a HEI phase with smaller lattice constants (Supplementary Fig. 1a). At 850 °C, the Pt (111) and Pt alloy (111) peaks merged at around 16.0° (Supplementary Fig. 1e). Rietveld refinement of HEI/KB (850 °C, 60 min) gave a low weighted profile $R$-factor ($R_{wp}$) of 2.4% (Supplementary Fig. 1f), suggesting a reliable fit and successful incorporation of all five elements into the HEA and HEI phases. Prolonged annealing is expected to promote complete conversion to the HEI phase.

Based on the structural evolution of high-entropy intermetallic NPs described above, two catalysts with approximate molar fractions of $Pt_4CoNiFeCu$ with and without N-dopants, namely $L1_0$-ordered $Pt_{51}Co_{11}Ni_{13}Fe_{12}Cu_{13}$ (HEI/KB) and N-doped $L1_0$-ordered $Pt_{48}Co_{10}Ni_{12}Fe_{12}Cu_{12}N_6$ (N-HEI/KB), were synthesized by optimizing the synthesis parameters (see details for synthesis in the "Methods" section and Supplementary Table 1). The atomic ratio of Pt element to total TMs in both HEI/KB and N-HEI/KB is close to 1:1, which is also confirmed by the inductively coupled plasma-mass spectroscopy (ICP-MS) results (Supplementary Table 2). The XRD results of these catalysts indicate a shift in the peak positions of N-HEI/KB towards lower angles compared to those of HEI/KB (Supplementary Fig. 1g), which suggests successful doping of N atoms with resultant tensile strain in HEI NPs[4]. The shoulder observed in Supplementary Fig. 1g can be attributed to small HEI particles with Pt-rich shells, supported by low $R_{wp}$ values for N-HEI/KB (4.8%) and HEI/KB (4.6%) using a two-phase model (cubic Pt and tetragonal HEI; Supplementary Fig. 1h, i). The slightly larger lattice parameter of the HEI phase in N-HEI/KB (3.680 ± 0.003 Å) compared to HEI/KB (3.675 ± 0.003 Å) confirms N-doping-induced strain (Supplementary Table 3). Additionally, similar HEI phase ratios in N-HEI/KB (54.1 wt%) and HEI/KB (59.4 wt%) suggest minimal structural impact from N-doping. Both the transmission electron microscopy (TEM) and high-angle annular dark-field scanning TEM (STEM-HAADF) images show uniform particle distributions for HEI/KB and N-HEI/KB, with average sizes of 2.6 ± 0.1 and 2.9 ± 0.1 nm, respectively (Supplementary Figs. 2a–c and 3a–c). Energy-dispersive spectroscopy (EDS) maps and line scans further validate the uniform distribution of all elements in N-HEI and HEI NPs (Supplementary Figs. 2d–f and 3d–f). A representative Z-contrast STEM-HAADF image of N-HEI/KB to atomic resolution displays clear periodic atomic columns with strong and weak image contrast, corresponding to Pt columns and TM columns (Fig. 1d and Supplementary Fig. 4a, b), characteristic of its intermetallic structure. The lattice spacing of 0.37 nm corresponds to the (001) plane of $L1_0$-ordered N-HEI (Fig. 1e), which is in agreement with XRD

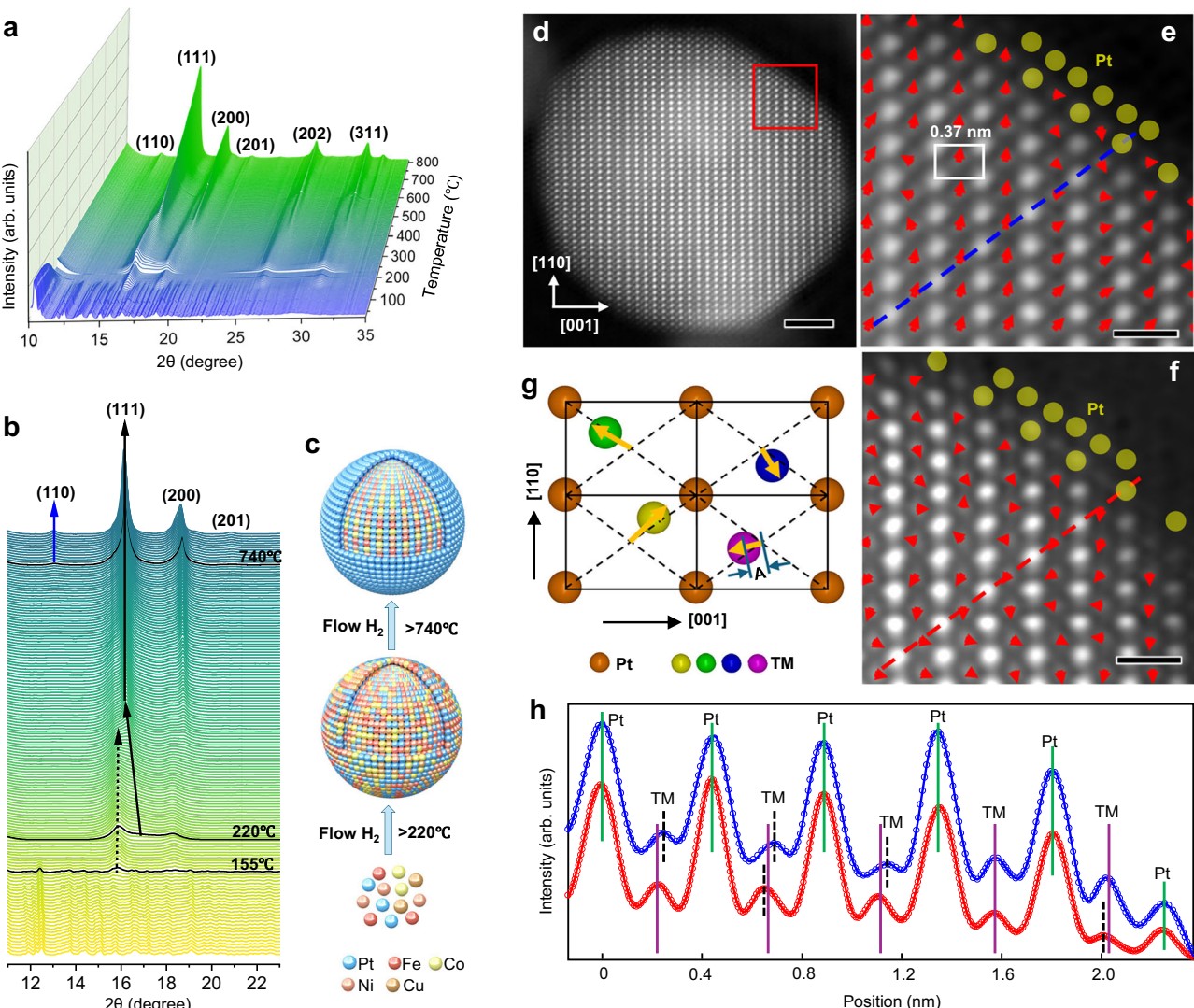

**Fig. 1 | Structure evolution and properties of the N-HEI/KB catalyst. a** and **b** In situ synchrotron XRD patterns for the structure evolution of HEI/KB during annealing in 10% $H_2$/He gas stream (wavelength 0.6199 Å). **c** Schematic for the synthesis of HEI/KB. **d** Representative atomic-resolution STEM-HAADF image of N-HEI particle viewed along [−110] direction. The image is lightly filtered in frequency space by applying a periodic mask to remove noise. The image contrast is approximately proportional to $Z^{1.7}$ along the atomic column; thus, the dots with strong and weak image contrast correspond to Pt and TM columns, respectively. **e** Enlarged image from the area marked by the red rectangle in (**d**). The white rectangle marks the lattice of N-HEI structure. **f** STEM-HAADF image from part of a HEI particle (see Supplementary Fig. 4b for the whole particle). The yellow spheres mark the Pt atoms at the edge. Scale bars are 2 nm for (**d**) and 0.5 nm for (**e**, **f**). **g** Schematic of the sub-angstrom displacement of TMs (transition metals). Four lattices are outlined by solid lines with Pt atoms at the apex. The arrows show the displacement of the TM from the center of the lattice (cross of the dashed lines) with their head and tail representing the direction and amplitude (*A*) of the displacement. **h** Intensity profiles (small circles) from scan lines in **e** and **f** with the same color (blue for N-HEI and red for HEI). The solid lines are refined curves using a Gaussian function. The green, purple, and black vertical lines mark the positions of Pt columns, the center of the lattice, and TM columns, respectively. The uncertainty of the HAADF TM position fitting is -0.02 Å. Source data for Fig. 1 are provided as a Source Data file.

measurements of 0.369 nm (Supplementary Fig. 1g), further affirms the formation of an ordered intermetallic structure.

Figure 1e provides an enlarged view of the STEM-HAADF image of the N-HEI NP (marked by a red rectangle in Fig. 1d), revealing a core-shell structure consisting of a high-entropy intermetallic core covered by a Pt monolayer (ML) shell. This could effectively inhibit the leaching of TM atoms from the surface of the catalysts, leading to improved stability. Figure 1e also reveals discernible atomic column displacements of TMs from the center of the lattice as indicated by the arrows where the head and tail of the arrow represent the direction and amplitude (*A*) of the displacement, respectively, as illustrated schematically in Fig. 1g. These displacements serve as evidence for the presence of sub-angstrom strain within the N-HEI core. Here, the TM column positions were determined by refining the peak position of TM

columns in the image using the second polynomial function, while the center of the lattice was determined by refining the positions of four Pt columns at the lattice apex. Interestingly, the displacement vectors exhibit randomness in both length and direction (Fig. 1e and Supplementary Fig. 4b), suggesting the coexistence of local tensile and compressive strains in the N-HEI core. The HEI catalyst has a similar structure to N-HEI, i.e., $L1_0$-orderd core–shell structure with a Pt ML shell, except that the HEI has no N atoms in the $L1_0$ structure (Fig. 1f and Supplementary Fig. 4c, d). The HEI catalyst also exhibits similar atomic displacements of TMs, characterized by random displacements (directions and amplitudes) as shown in Fig. 1f and Supplementary Fig. 4d. We think that the sub-angstrom tensile and compressive strains effectively inhibit dissolution of TM atoms from both N-HEI and HEI, ensuring their improved durability, as discussed later. As a

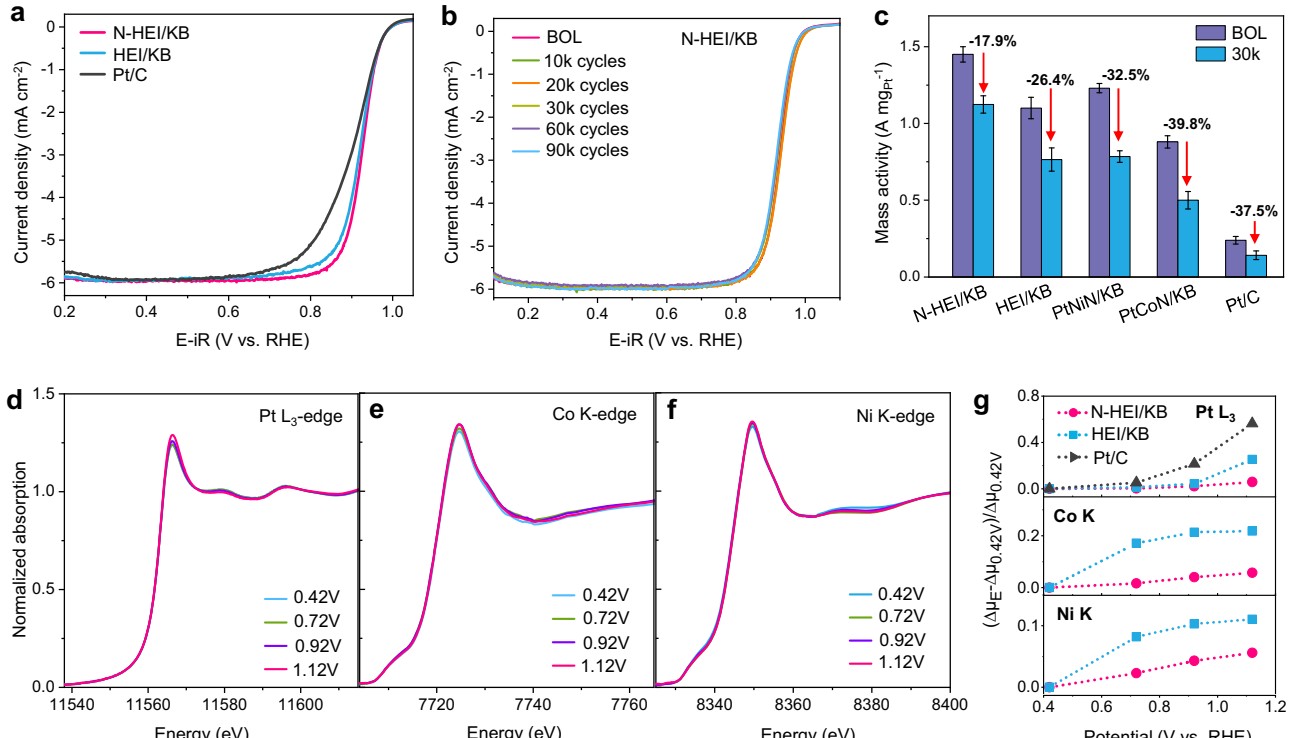

**Fig. 2 | Superior stability of the N-HEI/KB cathode. a** ORR polarization curves of N-HEI/KB, HEI/KB, and commercial Pt/C in $O_2$-saturated 0.1 M $HClO_4$ (pH = 1.0 ± 0.1) at 25 °C with a rotation rate of 1600 rpm and a scan rate of 10 mV $s^{-1}$. **b** ORR polarization curves of N-HEI/KB catalyst tested at the beginning of life (BOL) and different potential cycles with a scan rate of 10 mV $s^{-1}$ at 1600 rpm. **c** Mass activity (at 0.9 V) of N-HEI/KB, HEI/KB, PtNiN/KB, PtCoN/KB, and commercial Pt/C catalysts at BOL and 30k cycles. The error bars represent the standard deviation derived from three independent experimental measurements. **d**–**f** In situ XANES spectra of N-HEI/KB collected at different potentials in fluorescence mode at the **d** Pt $L_3$-edge, **e** Co K-edge, and **f** Ni K-edge. **g** Comparison of the adsorption edge peaks (Pt, Co, Ni) of the XANES spectra for N-HEI/KB, HEI/KB and Pt/C as a function of potential obtained in 0.1 M $HClO_4$ (relative to $\Delta\mu$ at 0.42 V). Source data for Fig. 2 are provided as a Source Data file.

comparison, there is basically no discernible amplitude in the displacement of TMs in the N-doped $L1_0$-ordered PtNi particle (PtNiN/KB, Supplementary Fig. 4e, f), further confirming the existence of the unique sub-angstrom strain in HEI structures. Figure 1h shows image intensity profiles along [111] direction, indicated by the blue line in Fig. 1e and the red line in Fig. 1f for single N-HEI and HEI NPs, respectively. It should be noted that we use a disordered fcc lattice (*a* = 3.7 Å) to index the crystalline direction and plane for consistency. The deviation of some TM atoms (black vertical dotted lines in Fig. 1h) from the center (purple lines) between two consecutive Pt atoms (green lines) is clearly seen, especially for N-HEI. Moreover, the STEM-HAADF image may be interpreted as the projected atomic potential along the beam direction convolved with the probe function. The peak width in the image intensity profile obtained from the STEM-HAADF image reflects the alignment of the atomic columns along the beam direction. When the atoms are misaligned along the beam direction due to the displacement, the peak width increases. Here, we use the full width at half maximum (FWHM) ratio, $FWHM_{TM}/FWHM_{Pt}$, in the image intensity profiles to quantify the average displacements of TMs along [111] direction for the N-HEI and HEI. Based on the refinement of intensity profiles, the average FWHM ratios of TM columns to Pt columns are 0.98 ± 0.08 and 0.88 ± 0.08 for N-HEI and HEI, respectively, indicating that a larger displacement is present in the N-HEI sample, which is a plausible origin of the improved durability of this catalyst.

More information on the electronic/structural states of the N-HEI and HEI catalysts can be obtained from the analysis of the synchrotron XAS spectra at hard and soft energy ranges. Compared to the HEI catalyst, the white line (WL) intensity of TMs (Co, Ni, Fe, and Cu) at the K-edges in the X-ray near-edge structure (XANES) spectra increases in N-HEI (Supplementary Fig. 5). Furthermore, the soft XAS spectra reveal

slight shifts of the L-edge features and WL peaks of TMs in N-HEI toward higher energy (Supplementary Fig. 6), compared to those in HEI. This change is probably a result of the formation of multiple TM-N bonds, where N atoms withdraw electrons from the TMs, leading to a reduced electron population in 3*d* band of TMs[31]. It is worth noting that the formation of Pt–N bonds is challenging due to the absence of the required conditions for their formation (~50 GPa and 2000 K)[32]. Consequently, we propose that the slightly increased WL intensity of Pt for the N-HEI compared to the HEI (Supplementary Fig. 5e) is likely a result of the enhanced d-band vacancy of the Pt[33], which is induced by the increased d-band vacancies of the TMs[34]. The Fourier transform spectra of the extended X-ray absorption fine structure (FT-EXAFS) at Pt $L_3$-edge reveal an expansion of Pt–Pt bonds in N-HEI compared to HEI (Supplementary Fig. 5f), indicating that N dopants induce tensile strain in Pt atoms. Combining these XAS findings with the high-resolution STEM analysis, N-doping further leads to the atomic displacement of TMs within the $L1_0$-ordered intermetallic structure due to the formation of TM–N bonds, which, in turn, introduces tensile strain in Pt atoms. These sub-angstrom strains, comprising both compressive and tensile components, may not only optimize the electronic structure of HEI catalysts to enhance catalytic activity but also effectively prevent the TM atoms from dissolving from the core during the ORR process, as discussed below.

## Electrochemical performance and structural stability

Electrochemical tests were performed to assess the ORR performance of N-HEI/KB under half-cell conditions, with comparisons made to HEI/KB and commercial Pt/C. The results, evaluated through half-wave potentials ($E_{1/2}$) from the polarization curves (Fig. 2a), establish an increase in ORR activity at the beginning-of-life (BOL) as follows: N-

HEI/KB > HEI/KB > commercial Pt/C, in which N-HEI/KB (925 mV) represents a 25 mV positive shift in comparison with the commercial Pt/C (900 mV). The electrochemical surface areas (ECSAs) obtained by the underpotential deposited hydrogen ($H_{upd}$) method for N-HEI/KB, HEI/KB, and commercial Pt/C were 62.1, 54.8, and 83.3 $m^2\,g_{Pt}^{-1}$, while those by the CO stripping were 77.6, 69.4, and 85.7 $m^2\,g_{Pt}^{-1}$, respectively (Supplementary Fig. 7 and Tables 4, 5). Notably, the ECSAs by the CO stripping for N-HEI/KB and HEI/KB are 1.27 times higher than those obtained via the $H_{upd}$ method, possibly due to suppression of hydrogen adsorption by sublayer transition metals[35,36]. Both HEI catalysts show an increase in mass activity and specific activity (MA and SA) for the ORR. Specifically, the N-HEI/KB catalyst exhibits the highest MA and SA of 1.45 $A\,mg_{Pt}^{-1}$ and 2.37 $mA\,cm^{-2}$ at 0.9 V (Supplementary Table 4), which are 6.0 and 8.2 times higher than those of the commercial Pt/C (0.24 $A\,mg_{Pt}^{-1}$ and 0.29 $mA\,cm^{-2}$), respectively (Supplementary Table 5). The HEI/KB showed 1.10 $A\,mg_{Pt}^{-1}$ and 2.01 $mA\,cm^{-2}$, respectively. The superior ORR performance of N-HEI/KB is further demonstrated by accelerated durability tests (ADT) involving cycling the potential between 0.6 and 0.95 V. As shown in Fig. 2b, N-HEI/KB exhibits minimal decay under different cycles, with only 5 mV decay of $E_{1/2}$ after 30,000 ADT cycles. Additionally, the MA of N-HEI/KB decreases by only 17.9% after 30,000 ADT cycles, which is much lower than those of HEI/KB (26.4%), PtNiN/KB (32.5%), PtCoN/KB (39.8%), and commercial Pt/C (37.5%), suggesting the superior durability of N-HEI/KB (Fig. 2c, Supplementary Figs. 7, 8 and Tables 4, 5). The ORR performance of N-HEI/KB is comparable to that of most Pt-based catalysts reported in recent literature (Supplementary Table 6). The increased MA and slightly decreased ECSA in the early cycles (10k and 20k cycles, as shown in Supplementary Fig. 7) are likely attributed to an increase in SA due to increasing Pt active sites, caused by the dissolution of the TM atoms from the surface of N-HEI/KB NPs and the consequent structural reconstruction on the catalyst surface, such as reducing low-coordinated sites. It is encouraging that both the ORR performance of N-HEI/KB and HEI/KB far exceeds the DOE target (0.44 $A\,mg_{Pt}^{-1}$ in MA and <40% MA loss after 30,000 cycles). It should be noted that even after 90,000 ADT cycles, the ORR performance of N-HEI/KB (0.98 $A\,mg_{Pt}^{-1}$ in MA and 33.1% in MA loss) still meets the DOE target.

The changes in morphology and composition of N-HEI/KB and HEI/KB are further assessed after the following stability tests. As shown in Supplementary Fig. 9, both N-HEI/KB and HEI/KB maintain their morphology after 30k ADT cycles, exhibiting no obvious NP loss or agglomeration. This highlights the robust structural stability of the HEI catalysts. In contrast, STEM images of PtNiN/KB, PtCoN/KB, and commercial Pt/C reveal clear particle agglomeration and an increase in particle size (Supplementary Fig. 10). Despite some inevitable dissolution of TM elements, EDS analysis shows that all elements in N-HEI/KB and HEI/KB are still uniformly distributed in the NP cores (Supplementary Fig. 9). In comparison, PtNiN/KB and PtCoN/KB exhibit greater TM dissolution (Supplementary Fig. 10 and Table 7), further emphasizing the superior compositional stability of the HEI structures. Combining the structural and functional analysis of the NPs, the improved performance of both N-HEI/KB and HEI/KB can be attributed to the formation of stable $L1_0$-ordered high-entropy intermetallic structures and the presence of sub-angstrom strain, which not only promotes catalytic activity but also prevents the TM dissolution to improve the durability of the catalyst.

The enhanced stability of the N-HEI/KB electrocatalyst is further evidenced by in situ XAS measurements under ORR conditions at various applied potentials. As shown in Fig. 2d–f, the XANES profiles for Pt $L_3$-edge, Co K-edge, and Ni K-edge in N-HEI/KB show minor increases in WL intensities with increasing potential, suggesting minimal oxidation state changes. Whereas Fe and Cu show more pronounced shifts in their XANES spectra toward higher energy (Supplementary Fig. S12), indicating greater sensitivity of these elements in the harsh acidic electrochemical environments. For clearer comparison, the relative changes in WL intensity ($(\Delta\mu_E - \Delta\mu_{0.42V})/\Delta\mu_{0.42V}$) (Pt, Co, and Ni) and the changes in half-step energy (energy at 0.5 of the normalized Fe and Cu XANES spectra) are depicted in Fig. 2g and Supplementary Fig. 12e, f. Across all TMs, N-HEI/KB shows reduced potential dependence compared to HEI/KB (Supplementary Fig. 11). This is probably due to the larger displacements present in N-HEI/KB, as they could act as an energy barrier for atomic diffusion of TMs (see the discussion in the DFT section, below). Furthermore, the fitting results of Pt $L_3$-edge spectra show that the Pt-Pt distance of the N-NEI/KB catalyst is 2.710 Å (Supplementary Fig. 13 and Table 8), which is longer than that of the HEI/KB catalyst (2.698 Å). We consider that the longer Pt-Pt distance in the N-NEI/KB catalyst could be caused by the enhanced strain in the TM layers due to N dopants. Additionally, the formation of multiple TM–N bonds can create a pinning effect of N atoms to TM atoms[4,31], which suppresses the dissolution/oxidization of the TMs, thereby enhancing the durability of the N-HEI/KB catalyst.

## MEA performance in HDV application

The enhanced stability of N-HEI/KB electrocatalyst is further demonstrated through MEA performance under HDV conditions. To verify the high practicability of N-HEI/KB in real HDV, we conducted the ADTs between 0.60 and 0.95 V with 0.20 $mg_{Pt}\,cm^{-2}$ for the cathode under $H_2$/air fuel cells. Operating at 250 $kPa_{abs}$ pressure and 100% relative humidity (RH), the N-HEI/KB cathode achieves a substantial current density of 1517 $mA\,cm^{-2}$ at 0.7 V at BOL (Fig. 3a and Supplementary Fig. 14), which is comparable to the commercial Pt/C (1520 $mA\,cm^{-2}$). After 90,000 voltage cycles, the current density of N-HEI/KB cathode still reaches 1388 $mA\,cm^{-2}$ at 0.7 V (Fig. 3b) and the voltage loss at 0.8 $A\,cm^{-2}$ is only 9 mV, demonstrating a competitive performance against most of the reported Pt-based intermetallic catalysts (Supplementary Table 9) but still meeting the latest DOE M2FCT target for PEMFCs in HDV applications (1300 $mA\,cm^{-2}$ at 0.7 V after 90k cycles)[30,37]. In contrast, the commercial Pt/C shows only 436 $mA\,cm^{-2}$ at 0.7 V for the same voltage cycles, demonstrating a rapid decline in activity with an increased number of ADT cycles (Fig. 3b, Supplementary Fig. 15 and Table 10). Most importantly, the MEA performance of N-HEI/KB remains virtually unchanged after 30,000 cycles at different RHs (Supplementary Fig. 14f), while the commercial Pt/C shows significant degradation with cycling and varying RHs (Supplementary Fig. 15e). This highlights the overall stability of the N-HEI/KB cathode in HDV applications.

We also conducted $H_2$–air polarization tests at different cathode loadings, as shown in Supplementary Fig. 16. The N-HEI/KB catalysts with cathode loadings of 0.10 and 0.05 $mg_{Pt}\,cm^{-2}$ exhibit higher MEA performance at BOL compared to a cathode loading of 0.2 $mg_{Pt}\,cm^{-2}$. This improvement is likely due to higher Pt utilization with lower cathode loading and enhanced mass transport of oxygen within the MEA. Even with reduced stability observed at lower cathode loadings, the N-HEI/KB cathode loaded with 0.05 $mg_{Pt}\,cm^{-2}$ maintains a high current density of 1493 $mA\,cm^{-2}$ at 0.7 V after 30,000 cycles. These findings underscore the significance of N-HEI/KB catalysts for HDV applications and highlight their robust performance for a range of different cathode loadings.

STEM-HAADF images of N-HEI/KB measured after MEA testing reveal that the N-HEI NPs remain dispersed across the carbon support after 90,000 voltage cycles (Supplementary Fig. 17), despite slight particle agglomeration and an increase in average particle size of 1.3 nm. After ADT cycles, the atomic-resolution STEM images clearly show the presence of a $L1_0$-ordered intermetallic phase with a well-defined core-shell structure in N-HEI (Fig. 3c and Supplementary Fig. 18). Elemental mapping of a representative N-HEI NP sample confirms that all the elements are maintained within the core structure (Supplementary Fig. 17c), and that a thicker Pt shell (3–7 atomic layers in thickness) surrounds the intermetallic core (Fig. 3c and Supplementary Fig. 17c), which contributes to the overall durability.

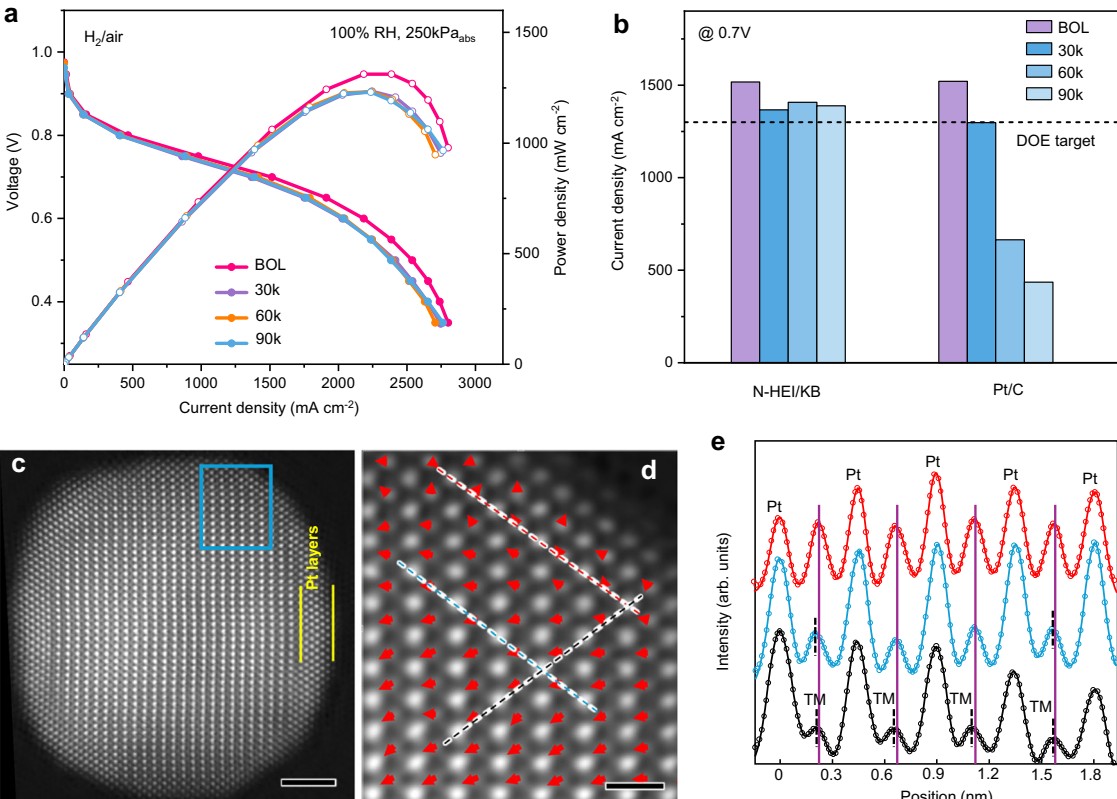

**Fig. 3 | Cycling and structural stability of the N-HEI/KB catalysts under HDV conditions. a** $H_2$/air fuel cell performance of N-HEI/KB at BOL and different voltage cycles, 0.20 $mg_{Pt}$ $cm^{-2}$ (cathode Pt loadings), $H_2$/air (500/2000 sccm), 80 °C, and 250 $kPa_{abs}$ pressure. The fuel cell resistance (6.65 ± 0.42 mΩ) was obtained from high-frequency resistance measurements. **b** Current density of N-HEI/KB and commercial Pt/C at 0.7 V under different ADT cycles. **c** Representative atomic-resolution STEM-HAADF image of N-HEI nanoparticle after 90k ADT cycles, showing multilayers in Pt shell. **d** Magnified image from the area marked by blue rectangle in **c** with TM displacement arrows embedded. Scale bars are 2 nm for (**c**) and 0.5 nm for (**d**). **e** Intensity profiles (dots) from scan lines in **d** with the same color. The solid lines are refined curves using a Gaussian function. The purple and black vertical lines mark the positions of the center of the lattice and TM columns, respectively. The TM displacement remains in the core (blue line and black line), but is suppressed in the Pt shell (red line). Source data for Fig. 3 are provided as a Source Data file.

Moreover, the atomic displacement of TMs is still evident in the N-HEI NP after ADT cycles, albeit slightly suppressed, as shown by the displacement arrow map in Fig. 3d and Supplementary Fig. 18a, b. The average FWHM ratio of TM columns to Pt columns is 0.95 ± 0.03 (Fig. 3e), which closely resembles the BOL value of 0.98. Similarly, atomic displacement of TMs in N-HEI before and after ADT cycles indicates that the introduced strain in N-HEI is very stable during the MEA testing, which effectively suppresses the dissolution of TMs from the $L1_0$-ordered core, thereby ensuring superior durability.

## DFT calculations and uMLIPs results

DFT calculations and uMLIPs methods were independently conducted to further elucidate the origin of enhanced ORR performance of N-HEI, with details provided in Methods and Supplementary Information. To understand the tendency towards short-range ordering (SRO) in N-HEI with the $L1_0$ phase, we first analyzed the cohesive energies of all 25 binary (AB) combinations of Co, Ni, Fe, Cu, and Pt within $L1_0$-AB intermetallic structures. The uMLIPs and DFT results, presented in Fig. 4a and Supplementary Fig. 19, are consistent, showing that Cu–Cu pairs have the highest cohesive energies ($E_{coh}$(Cu-Cu) = −3.5 eV/atom) across all AB pairs, this suggests preferential formation of localized atomic Cu clusters within the $L1_0$-AB and $L1_0$-HEI structures. This finding aligns with previous studies by Chen et al. on $L1_2$-AB$_3$ and $L1_2$-HEI structures involving Co, Cu, Fe, Ni, Pd, and Ti[22]. Their work suggests that the disparity and exclusivity in chemical affinity are the primary factors driving the microstructure of high-entropy disordered systems, especially when the differences in atomic size and electronegativity among the constituent elements are minimal. Given the similar atomic size and electronegativity between Pt and Pd, we hypothesize that similar principles apply to Co/Ni/Fe/Cu/Pt in $L1_0$-HEIs. Additionally, Pt-containing pairs constantly have lower cohesive energies than other pairs, implying a tendency for Pt-containing pair formation without preferential choice of pair partners in $L1_0$-HEI catalysts. Based on this binary interatomic system, we investigated the SRO in $L1_0$-HEIs through large-scale atomic modeling with fine-tuned uMLIPs[38]. The efficiency and accuracy of fine-tuned uMLIPs in comparison with the DFT results are discussed in the Supplementary Figs. 20, 21. To account for the configurational entropy and local lattice distortions inherent to HEIs, it is essential to consider a wide range of multiple-site configurations. Accordingly, we randomly generated and relaxed 500 HEI structures of equimolar compositions (each containing 144 atoms), as illustrated in Fig. 4c and Supplementary Fig. 22, to comprehensively sample the configuration space. We evaluated the SRO of these HEI structures by calculating the relative pair probability (RPP) parameters for all Co/Ni/Fe/Cu atom pairs[39]. The RPP parameters represent the relative probability of finding two species for 1st, 2nd, and 3rd nearest neighbor (NN) pairs. As shown in Fig. 4b and Supplementary Fig. 23, the most pronounced deviation from a random arrangement was observed around the Cu and Fe atoms. Cu and Fe tend to pair with themselves (Cu–Cu and Fe–Fe) ($RPP^{1st}_{Cu-Cu}$ = 1.31 and $RPP^{1st}_{Fe-Fe}$ = 1.24 in the 1st NN pairs), while they separate from each other ($RPP^{3rd}_{Cu-Fe}$ = 1.36 in the 3rd NN pairs). Cu and Fe have a positive association with Ni and Co ($RPP^{1st}_{Cu-Ni}$ = 1.19 and $RPP^{1st}_{Fe-Co}$ = 1.33) in the 1st NN pairs, respectively. These observations are confirmed by a

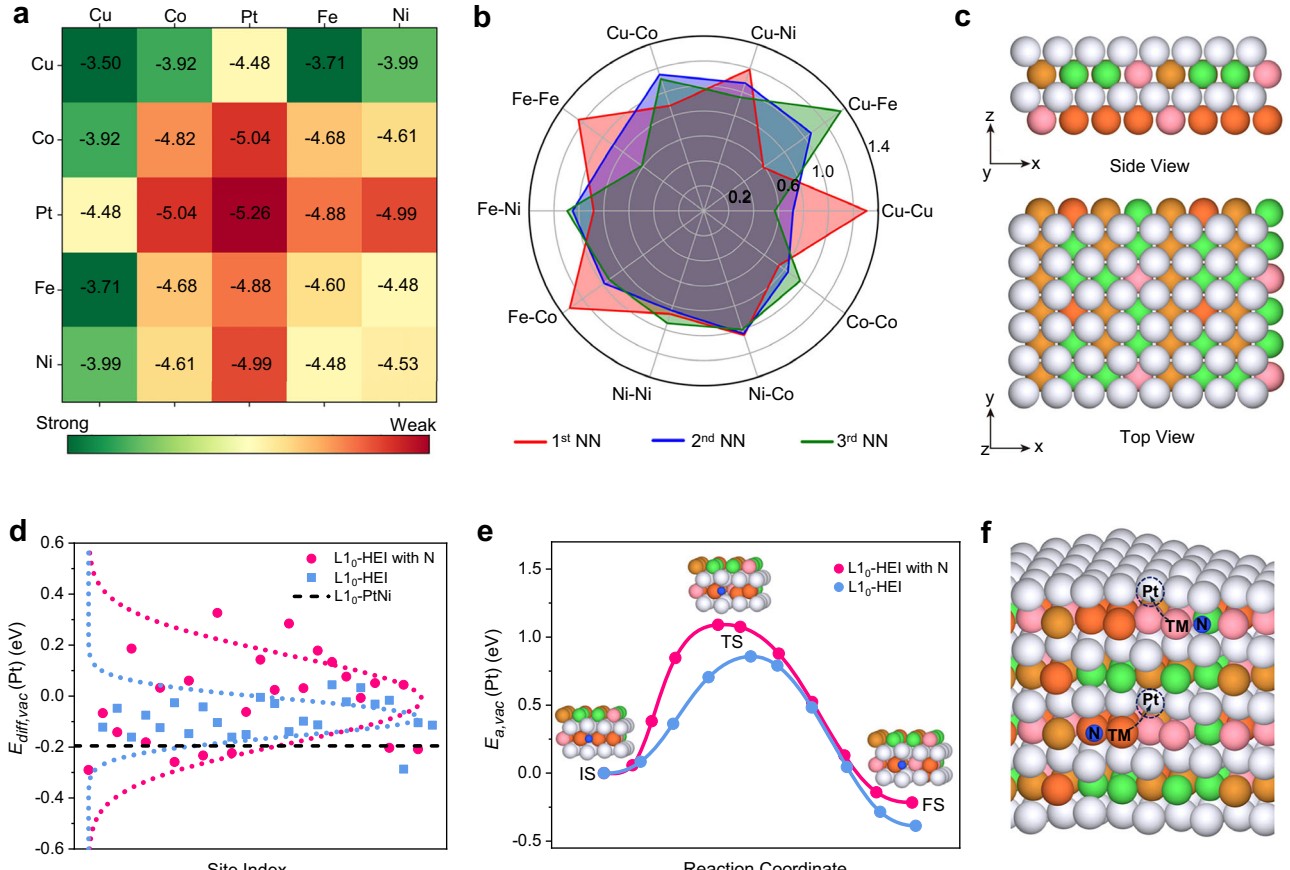

**Fig. 4 | DFT and uMLIPs calculations of the N-HEI nanoparticle. a** Cohesive energy of the $L1_0$ unit cell calculated by fine-tuned universal Machine Learning Interatomic Potentials (uMLIPs). **b** The average relative pair probability (RPP) parameters for Co, Fe, Ni, and Cu elements up to the 3rd NN shell. RPP = 1 indicates that the two species are as likely to be found within a given neighboring shell as any atom pairs. RPP > 1 (<1) indicates that they are more (less) likely to be found in that neighboring shell. **c** Side and top view of the $L1_0$-HEI model (Pt: gray, Co: pink,

Ni: green, Fe: orange; Cu: brown). **d** Pt vacancy diffusion energy of $L1_0$-HEI with/without N-doping and $L1_0$-PtNi using uMLIPs methods. **e** The diffusion barrier path and activation energy of Pt vacancy for $L1_0$-HEI with/without N-doping using DFT calculations (IS: initial state, TS: transition state, FS: final state). **f** Schematic diagram of Pt-vacancy-mediated diffusion of TM atoms in the $L1_0$-HEI with N slab model. Source data for Fig. 4 are provided as a Source Data file.

combination of molecular dynamics and Monte Carlo simulations of Co/Ni/Fe/Cu/Pt $L1_2$-HEI[22], which further supports an increased likelihood of Cu and Fe atomic clustering in $L1_0$-HEIs. Consequently, the theoretical findings suggest that the sub-angstrom strain in Co/Ni/Fe/Cu/Pt $L1_0$-HEIs could be associated with the localized segregation of Cu and Fe atoms within the TM layers.

To understand the strain effect in N-HEI on ORR stability, we explored the diffusion energy of Pt vacancies in the most stable $L1_0$-HEI models with/without N-doping by uMLIPs and compared these results to binary $L1_0$-PtNi. We first evaluated the formation energies of various N interstitial configurations and Pt vacancies in $L1_0$-HEI and $L1_0$-PtNi structures (Supplementary Fig. 24a and b, respectively). The results in Supplementary Fig. 24c demonstrate that N interstitials have lower formation energies in $L1_0$-HEI (1.42 eV for the average in 24 sites) than in $L1_0$-PtNi (1.88 eV), indicating that N-doping is easier in $L1_0$-HEI compared to $L1_0$-PtNi. Moreover, $L1_0$-HEI has higher formation energies of Pt vacancies than $L1_0$-PtNi (Supplementary Fig. 24d), pointing to an enhanced structural stability of $L1_0$-HEI, due potentially to the localized strain from the atomic clustering of Cu and Fe. Among $L1_0$-PtNi, $L1_0$-HEI, and $L1_0$-N-HEI structures, $L1_0$-N-HEI showed the highest diffusion energy of Pt vacancies, followed by $L1_0$-HEI (Fig. 4d), indicating that the HEI structure impedes the diffusion of Pt vacancies compared to the PtNi binary structure, and N-doping can further suppress it. Additionally, we present the mapping plot of site-dependent local atomic distortion in the Pt and TM layers of $L1_0$-N-HEI, $L1_0$-HEI, and $L1_0$-PtNi (Supplementary

Fig. 25). The local atomic distortion in the $L1_0$-HEI ($\Delta r_i = 0.037$ Å) is significantly greater than in $L1_0$-PtNi ($\Delta r_i = 0.002$ Å). After N-doping, more atomic distortion is observed in the $L1_0$-N-HEI ($\Delta r_i = 0.092$ Å), which increases the energy barrier for atomic activation and migration, thus reducing the diffusion rate of atoms.

The energy barriers for the diffusion path and activation energy of Pt vacancies were further verified using DFT calculations with the climbing image nudged elastic band (CI-NEB) method[40]. The Pt vacancy neighboring the TM atoms is built into the diffusion model (Fig. 4f), and the results are shown in Fig. 4e. The energy barrier for diffusion of TM atoms to the Pt vacancies increases from 0.86 to 1.09 eV after N-doping, suggesting that N dopants impede the diffusion of TMs in HEI. Therefore, the leaching of TMs during long-term ORR operation could be effectively slowed, leading to the improved stability of N-HEI. This could be attributed to the localized strain caused by the TM displacement and the pinning effect arising from the formation of metal-N bonds. On the other hand, high diffusion energy barriers indicate a diminished propensity for Pt vacancy segregation, thereby enhancing the stability of N-doped HEI. These results corroborate the origin of the improved ORR stability of N-HEI and align with our experimental observations.

## Discussion

In summary, we synthesized an N-doped $L1_0$-ordered HEI electrocatalyst, which exhibited high ORR electrocatalytic activity and

stability in both RDE and MEA testing for HDV applications. Detailed experimental measurements in combination with the DFT calculations and uMLIPs suggest that the origin of sub-angstrom strain in the N-HEI structure is likely attributed to local atomic segregation and the effects of interstitial N dopants. The high performance of N-HEI/KB originates from the synergy between the sub-angstrom strain effect, the pinning effect of TM−N bonds, and the high-entropy effect, which collectively enhance the corrosion resistance of the catalysts. This work not only offers an effective and practical strategy to improve the structural stability of electrocatalysts for fuel cell applications, but also caters to immediate market needs while laying the groundwork for broader adoption, particularly in the real realm of zero-emission heavy-duty transportation.

## Methods

### Materials and chemicals
The Ketjenblack (KB) EC-600JD carbon was bought from Fuel Cell Store. Platinum (II) acetylacetonate ($Pt(acac)_2$, 98%), nickel (II) acetylacetonate ($Ni(acac)_2$, 95%), cobalt (II) acetylacetonate ($Co(acac)_2$, 97%), iron (III) acetylacetonate ($Fe(acac)_3$, 97%), and copper (II) acetylacetonate ($Cu(acac)_2$, 97%) were obtained from Sigma-Aldrich (USA). Acetone ($CH_3OCH_3$, HPLC, ≥99.5%) and iso-propanol (IPA, HPLC, 99.9%) were purchased from Spectrum Chemical MFG Group and Sigma-Aldrich (USA), respectively. All aqueous solutions and suspensions were prepared using Millipore ultrapure water (18.2 MΩ cm). Gases, including Ar (99.999%), $O_2$ (99.6%), $H_2$ (99%, extra dry) and anhydrous ammonia ($NH_3$, 99.99% purity) were supplied by Praxair Inc. Commercial Pt/C (TEC10E50E, 46.4 wt% Pt) was purchased from Tanaka Kikinzoku Kogyo (TKK) Corporation. All reagents were used as received without further purification.

### Synthesis
A mixture containing 1 mmol $Pt(acac)_2$, 0.25 mmol each of $Co(acac)_2$, $Ni(acac)_2$, $Fe(acac)_3$, and $Cu(acac)_2$ was dissolved in 150 mL of acetone containing 600 mg of KB EC-600JD carbon. The resulting solution was subjected to ultrasonication for 2 h and then stirred magnetically overnight to achieve uniform dispersion of metal precursors. The solvent was then removed using a rotating evaporator device, resulting in a dry precursor powder. This powder was subjected to thermal annealing at 800 °C for 2 h at a ramp rate of 10 °C min$^{-1}$ under a flowing $H_2$/Ar atmosphere (10 vol% $H_2$), resulting in the formation of the PtFeCoNiCu intermetallic catalysts (named as HEI/KB). Second, the HEI/KB powder was further annealed in $NH_3$ gas at 500 °C for 2 h to obtain N-doped HEI (named as N-HEI/KB).

For comparison, bimetallic precursors of PtNi and PtCo were prepared using the same procedure by mixing 1 mmol $Pt(acac)_2$ with 1 mmol $Ni(acac)_2$ or 1 mmol $Co(acac)_2$, respectively, following the same dispersion and drying procedures. The resulting dried precursors were then annealed at 600 °C for 5 h (10 °C min$^{-1}$) in a tube furnace under $NH_3$. This process yielded N-doped PtNi and PtCo intermetallic catalysts, designated as PtNiN/KB and PtCoN/KB, respectively. Both PtNiN/KB and PtCoN/KB exhibit the L1$_0$-ordered structure, as confirmed by XRD analysis (Supplementary Fig. 8a, b).

### Characterization
STEM-HAADF imaging and EDS elemental mapping were performed using a FEI Talos F200X microscope equipped with a four-quadrant, 0.9-sr solid-angle EDS detector at an accelerating voltage of 200 kV in Brookhaven National Laboratory (BNL). High-resolution STEM-HAADF imaging was conducted using a double aberration-corrected JEOL-ARM200CF microscope with a cold-field emission gun and operated at 200 kV. The microscope is equipped with a JEOL HAADF detector with a collection angle of 67–275 mrad for incoherent HAADF (Z-contrast) imaging. XRD patterns were collected on a Rigaku SmartLab Universal Diffractometer with Cu-Kα radiation. The ICP-MS measurements were made with a PerkinElmer NexION 2000 ICP Mass Spectrometer calibrated using standards prepared from NIST traceable solutions.

### Electrochemical measurements
All electrochemical ORR measurements were conducted at room temperature using a conventional three-electrode setup connected to a Bio-Logic VSP potentiostat (Bio-Logic Science Instruments, France). A glassy carbon rotating disk electrode (GC-RDE, 5 mm diameter, 0.196 cm²) served as the working electrode. Prior to use, the GC electrode was carefully polished using 0.3 μm followed by 0.05 μm alumina slurry to achieve a clean and reflective surface. A Pt wire served as the counter electrode, while an Ag/AgCl electrode (3 M KCl) was employed as the reference. The catalyst ink was prepared by ultrasonically dispersing 5 mg of the catalyst into 5 mL of mixed solution (4 mL Milli-Q water, 1 mL isopropanol, and 5 μL 10 wt% Nafion). Subsequently, 10 μL of the ink was dropped onto the GC-RDE surface and allowed to dry in ambient conditions. The Pt loading of all samples was calculated from the Pt mass in the catalyst and the deposited ink volume, resulting in a loading of approximately 12 μg$_{Pt}$ cm$_{GC}^{-2}$. This value was used to determine the mass activity and the ECSA. The electrolyte (0.1 M $HClO_4$) was prepared freshly on the day of each experiment by diluting high-purity $HClO_4$ (65–71%, trace metal grade, Thermo Fisher Scientific) with Milli-Q water (18.2 MΩ cm). The solution was stored in a clean, airtight volumetric flask at room temperature and used within 24 h to ensure consistency and minimize contamination. The pH of the 0.1 M $HClO_4$ electrolyte was measured to be 1.0 ± 0.1 at room temperature (25 °C) using a calibrated pH meter. All measured potentials were converted to the reversible hydrogen electrode (RHE) using Eq. (1):

$$E_{RHE} = E_{Ag/Ag} + 0.197 + 0.059 \times pH \qquad (1)$$

The ECSAs of the catalysts at BOL were determined using both $H_{upd}$ and CO stripping methods. The CV measurements were performed in an Ar-saturated 0.1 M $HClO_4$ solution. To clean and stabilize the catalysts, 20 CV scans were first conducted at a scan rate of 100 mV s$^{-1}$. Subsequent measurements were performed at 20 mV s$^{-1}$ to determine the ECSA. The ECSA from the $H_{upd}$ method was calculated using Eq. (2):

$$ECSA = \frac{Q_H}{0.21 \times m_{Pt}} \qquad (2)$$

where $Q_H$ (in mC) represents the average charge of hydrogen adsorption and desorption, 0.21 mC cm$^{-2}$ is the charge required to oxidize a monolayer of hydrogen on Pt, and $m_{Pt}$ is the mass of Pt on the electrode (in mg).

The CO stripping test was conducted by bubbling CO gas into a 0.1 M $HClO_4$ electrolyte for 30 min while maintaining the electrode potential at 0.05 V. Subsequently, Ar gas was purged into the electrolyte for an additional 30 min to remove dissolved CO in the electrolyte. CV measurements were then performed by scanning from 0.05 to 1.05 V at a scanning rate of 50 mV s$^{-1}$. The ECSA from the CO stripping was calculated using Eq. (3):

$$ECSA = \frac{S_{CO}}{0.42 \times V_{scan} \times m_{Pt}} \qquad (3)$$

where $S_{CO}$, $V_{scan}$, and $m_{Pt}$ represent the integral area of CO stripping peak (in mA V), the scanning rate (in mV s$^{-1}$), and the Pt loading (in μg), respectively. We used an area-specific charge of two-electron transfer of 0.42 mC cm$^{-2}$. In the analysis of specific activity, we employed the ECSA values determined by the $H_{upd}$ method.

The LSV for the ORR was performed in an $O_2$-purged 0.1 M $HClO_4$ solution. Prior to measurement, the electrode was stabilized by cycling

between 0 and 1.1 V at 100 mV s$^{-1}$. LSV was then measured over the same potential limit using a sweep rate of 10 mV s$^{-1}$, with iR correction applied, at a rotation rate of 1600 rpm. Stability tests for the catalyst were carried out in air-saturated 0.1 M HClO$_4$ by potential cycling between 0.6 and 0.95 V, holding each potential for 3 s, at a scan rate of 100 mV s$^{-1}$ and room temperature. For comparative analysis, HEI/KB, PtNiN/KB, PtCoN/KB, and commercial Pt/C catalysts were evaluated under the identical experimental conditions. Electrochemical data were collected using the potentiostat software (EC-Lab V10.40) and further processed with Origin software. Each electrochemical measurement was repeated a minimum of three times. We selected three ORR curves to calculate error bars (standard deviation, SD) for mass activity, and the electrochemical curves (LSV and CV) presented in this work correspond to the median results.

## MEA fabrication

The as-synthesized catalysts were incorporated into MEAs using the catalyst-coated membrane (CCM) technique. Catalyst ink was directly deposited onto the surface of a 5 × 5 cm$^2$ proton exchange membrane (Gore® 8 µm, USA, untreated) using a spray system (Sono-tek®, USA). For the anode, a catalyst layer with 0.1 mg$_{Pt}$ cm$^{-2}$ Pt loading was applied using a commercial 30 wt% Pt-XC72 catalyst (JP30S®, China). The ink formulation consisted of the catalyst and ionomer (25 wt%, Aquivion® D-79-25BS) dispersed in a solvent mixture of n-propanol and DI water (volume ratio 1:6). The ionomer-to-carbon (I/C) weight ratio was set at 0.45. For the HDV application, the cathode catalyst layer was prepared with a Pt loading of 0.2 mg$_{Pt}$ cm$^{-2}$ and an I/C ratio of 0.55, while all other conditions were kept identical to those of the anode catalyst layer. The final MEA had an active geometric area of 4.98 cm$^2$ (3.51 cm × 1.42 cm). Non-woven carbon paper (GDL, 22BB Sigracet®, Germany) was used as the gas diffusion layer for both anode and cathode. The MEA was assembled in a sandwich configuration between two graphite plates (Supplementary Fig. 14a). A differential cell incorporating 14 parallel flow channels was employed to effectively reduce the pressure drop across the electrode and ensure consistent backpressure, even under high-flow rate conditions.

## MEA testing protocols

The MEAs were evaluated using a fuel cell testing station (Fuel Cell Technologies Inc., USA) following the U.S. DOE protocols. The MEAs were activated (break-in) for a 16-h period under H$_2$ (anode) and air (cathode) flows of 200 standard cubic centimeter per minute (sccm) and 400 sccm, respectively, at a cell temperature of 80 °C and a pressure of 150 kPa$_{abs}$. The activation involved scanning the cell voltage between 0.7 V and 0.35 V in 50 mV steps, holding for 5 min at each step. The polarization curves under H$_2$–air conditions were obtained by scanning the cell voltage from 0.35 V to open circuit voltage (OCV) in 50 mV increments, with 60 s held at each point. H$_2$ (anode) and air (cathode) flow rates were set at 500 and 2000 sccm, respectively, at a cell temperature of 80 °C and a pressure of 250 kPa$_{abs}$ under 100% RH. The corresponding current density was calculated by averaging the values in the last 15 s of each hold. High-frequency resistance (HFR) was determined at each point under 10 kHz. Catalyst MA was assessed at 80 °C and 100% RH under a pressure of 150 kPa$_{abs}$ with H$_2$ (anode) and O$_2$ (cathode) flow rates of 500 and 2000 sccm, respectively. The MA measurements were performed by holding the cell voltage at 0.6 V for 5 min, followed by 15 min at 0.9 V$_{iR-Free}$. The average current in the final 5 min (collected every second) was used to calculate MA with corrections applied for H$_2$ crossover. MEA stability was evaluated using a trapezoidal wave protocol cycling between 0.6 and 0.95 V, with a rise time of 0.5 s and hold time of 2.5 s at 100 kPa$_{abs}$, 80 °C, 100% RH, and H$_2$/N$_2$ flows of 50/75 sccm (anode/cathode). The fuel cell performance was evaluated every 30,000 cycles. The voltage reported in the I–V polarization curve represents the full-cell voltage. Polarization curve measurements for each MEA testing were performed once.

## In situ synchrotron XRD measurements

In situ synchrotron XRD experiments were conducted using an annealing cell at the QAS beamline (7-BM) of the National Synchrotron Light Source II (NSLS II) at BNL. Diffraction patterns were recorded with a large-area amorphous silicon detector (PerkinElmer 1621), featuring 2048 × 2048 pixels and a pixel size of 200 × 200 mm. Detector calibration was carried out using a LaB$_6$ standard to determine the sample-to-detector distance. The X-ray beam had a wavelength of 0.6199 Å. For the in situ heat-treatment tests, a mixture of Pt, Co, Ni, Fe, and Cu precursors supported on KB EC-600JD carbon (30 wt% total metal content) was used. The samples were heated from ambient temperature to 850 °C at a ramp rate of 10 °C min$^{-1}$ under a reducing atmosphere composed of 10 vol% H$_2$ and 90 vol% He.

## In situ XAS measurements

Hard XAS measurements, including both in situ and ex situ measurements, were performed using an electrochemical cell at the QAS beamline (7-BM) of the NSLS II at BNL. Energy tuning of the incident X-ray was achieved through a continuously scanning Si(111) crystal monochromator, enabling high-resolution selection of photon energies for XAS analysis. To suppress higher-order harmonic contaminations, the monochromator was detuned to reduce the incident X-ray intensity. The home-made in situ electrochemical cell (shown in Supplementary Fig. 11a [41]) consisted of a catalyst sample (working electrode), a proton exchange membrane (Nafion 117, DuPont Chemical Co., DE), and a Pt thin foil (counter electrode), which were tightly clamped between two acrylic plastic bodies with X-ray windows. A stationary, aerated 0.1 M HClO$_4$ solution was used as the electrolyte, and a leak-free Ag/AgCl electrode served as the reference electrode. XAS data analysis was performed using the Athena and Artemis software [42]. We performed the fitting of Pt L$_3$-edge spectra by lumping Co, Ni, Fe, and Cu as one TM. We used Ni as the representative TM of Pt-TM fitting in FEFF calculation within Artemis.

## Soft XAS measurements

We performed soft XAS measurements of Co L-edge, Ni L-edge, Fe L-edge, and Cu L-edge at room temperature at the IOS (23-ID-2) beamline of NSLS-II, BNL. Spectra were acquired under ultrahigh-vacuum conditions with a base pressure of 10$^{-9}$ Torr using both total electron yield (TEY) via drain current measurements and partial fluorescence yield (PFY) modes.

## DFT calculations

DFT calculations were carried out using the open-source GPAW software package, which implements the projector-augmented-wave (PAW) method [43,44]. Geometry optimizations and calculation analysis were managed through the atomic simulation environment (ASE) [45]. The Perdew–Burke–Ernzerhof (PBE) [46] functional was used to account for exchange and correlation energies. The calculations employed a plane-wave basis set with a 500 eV energy cutoff and a Fermi-level smearing width of 0.05 eV. Structural relaxations were performed by optimizing both atomic positions and lattice parameters of models until all the maximum atomic forces were <0.05 eV Å$^{-1}$. The Brillouin zone sampling used a Monkhorst–Pack scheme with an interval between k-points along reciprocal lattice vectors 0.08 π Å$^{-1}$. To investigate the structural and electronic properties, 25 binary combinations of Co, Ni, Fe, Cu, and Pt were constructed in the binary L1$_0$-AB inter-metallic configuration by alternately stacking TMs (TM = Co, Ni, Fe, Cu) and Pt atoms along the c-axis of the fcc structure (Supplementary Fig. 19). To generate datasets for the fine-tuned uMLIPs, we consider serials of structures inside the five-element HEI space to enable the prediction of arbitrary L1$_0$-HEI model (Supplementary Fig. 20). These comprehensive datasets use DFT simulations of structures covering pure and binary to quinary compositions, with and without nitrogen interstitials, and various supercell sizes. The diversity of configurations

was designed to capture a wide range of possible multi-site arrangements within the HEI system. These uMLIPs enable rapid and effective evaluations of the potential energy surface of HEIs for a given composition. To closely reflect experimental observations, an average composition of $Co_{0.25}Ni_{0.25}Fe_{0.25}Cu_{0.25}Pt$ was selected as the representative $L1_0$-HEI model. Using the fine-tuned uMLIPs, 500 randomly generated configurations were screened to identify the $L1_0$-HEI structure with the lowest potential energy, which was then used to calculate Pt vacancy formation energies and diffusion barriers, both in the presence and absence of nitrogen interstitials, as shown in Supplementary Fig. 24 and Supplementary Data 1. The vacancy formation energy $E_{vac}$ was calculated using Eq. (4):

$$E_{vac} = E_{[vac]} + E_{[M]} - E_{[pristine]} \qquad (4)$$

where $E_{[vac]}$ is the energy of $L1_0$-HEI system with a Pt vacancy, $E_{[M]}$ is the energy of a single Pt atom in its bulk, and $E_{[pristine]}$ is the energy of the corresponding $L1_0$-HEI system without any defects. The climbing image nudged elastic band (CI-NEB) method[40] was used to determine the minimum energy pathways for the diffusion of Pt vacancy. Nine and eight images were chosen to derive smooth potential energy curves for $L1_0$-HEI systems with and without nitrogen interstitials.

### Fine-tuning of uMLIPs and RPP parameters

In this study, we extrapolated the pre-trained M3GNet uMLIPs[38] to the PtCoNiFeCu $L1_0$-HEI system through a fine-tuning process. The pre-training dataset of M3GNet uMLIPs was primarily drawn from Materials Project ionic relaxation trajectories, and is therefore inherently limited in its exposure to diverse atomic configurations. As a result, this limitation makes it difficult to accurately describe the energy landscapes associated with out-of-domain (OOD) states like the $L1_0$-HEI structures. Additionally, as the specific settings of DFT calculations used in the pre-training datasets may differ from those used in this work, such discrepancies further necessitate fine-tuning to ensure consistent results with our DFT calculations. First, we generated a comprehensive structure dataset using DFT calculations (Supplementary Fig. 20). This dataset includes pure transition metals (5 structures), $L1_0$-AB intermetallic structures (15 structures), HEA structures (100 structures), $(2 \times 2 \times 4)$ supercell of $L1_0$-HEI intermetallic structures with/without N interstitial (200 structures), and $(2 \times 3 \times 4)$ supercell of $L1_0$-HEI intermetallic (50 structures). In total, there are 370 structures, including 11,738 intermediate structures. The dataset was divided into training and validation sets in a 9:1 ratio. The test set was taken from another DFT calculated $4 \times 3 \times 4$ supercell of $L1_0$-HEI (10 structures), containing 48 atoms. For the fine-tuning of uMLIPs, the models were trained in energy, force, and stress labels with loss weights of 1.0, 1.0, and 0.1, respectively, using the mean squared error (MSE) loss criterion. The Adam optimizer was used for optimization at a learning rate of $1e^{-4}$ for 200 epochs. The model checkpoint with the best force validation MAE was selected for test set predictions. The performance of fine-tuned models in MAE with increasing training steps on train and validation sets of the dataset is shown in Supplementary Fig. 21. The test MAE for single point calculation of initial structure and for relaxation to the final structure are shown in Supplementary Tables 11 and 12, respectively. The fine-tuned uMLIPs were used to predict the ground state configurations of the 500 randomly generated $4 \times 3 \times 4$ supercells of $L1_0$-HEI structures with an average composition of $Co_{0.25}Ni_{0.25}Fe_{0.25}Cu_{0.25}Pt$, each containing 48 atoms, as illustrated in Supplementary Fig. 22. This dataset encompasses a wide variety of distinct local environments, enabling comprehensive sampling of the configurational landscape of HEIs. By incorporating a diverse range of physically relevant multi-site configurations, the dataset ensures an accurate representation of the structural complexity inherent to HEI systems. The ionic relaxations were converged to a maximum interatomic force criterion of 0.05 eV $Å^{-1}$ for fine-tuned uMLIPs. On average,

the fine-tune uMLIPs relaxation takes 29.4 s on an Intel Core i5-10400 Processor 2.9 GHz 6 cores CPU machine and 10.8 s on a NVIDIA GeForce RTX3090 GPU machine, respectively. In comparison, DFT relaxations require 11,334.4 s on an Intel Xeon Platinum 8375 C 2.9 GHz 32-core CPU machine. The relative pair probability parameters[39], expressed as Eq. (5):

$$RPP_{\Delta r}(A, B) = \frac{p_{\Delta r}(A, B)}{p_{\Delta r}(\star, \star)} \frac{\rho^2}{\rho_A \rho_B} \qquad (5)$$

computes the number of pairs between TM A and TM B that occur within a range $\Delta r$ of distances, divided by the number of all TM pairs found in that same region, and normalized by the number density of the two species, $\rho_A$ and $\rho_B$ and the overall number density of TMs.

## Data availability

The data generated in this study are provided in the Supplementary Information/Source Data file. Source data are provided with this paper.

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

## Acknowledgements

This manuscript has been authored by employees/guests of Brookhaven Science Associates, LLC under contract no. DESC0012704 with the U.S. DOE. The publisher, by accepting this manuscript for publication, acknowledges that the United States Government retains a non-exclusive, paid-up, irrevocable, worldwide license to publish or reproduce the published form of this manuscript, or allow others to do so, for United States Government purposes. Additional work was performed by the Million Mile Fuel Cell Truck (M2FCT) Consortium, technology manager Greg Kleen, which is supported by the U.S. DOE, Office of Energy Efficiency and Renewable Energy, Hydrogen and Fuel Cell Technologies Office, under contract no. DE-SC0012704 (K.S., X.Z., H.C., X.B.). This work was also supported in part by the U.S. DOE, Division of Chemical Sciences, Geosciences and Biosciences, Office of Basic Energy Sciences and carried out at Brookhaven National Laboratory, operated under contract DE-SC0012704 (FWP-CO60). The research used the beamline 7-BM (QAS) and 23-ID-2 (IOS) at NSLS II, U.S. DOE, Office of Science User facilities. Work at the beamlines was supported in part by the Synchrotron Catalysis Consortium (SCC), U. S. Department of Energy Grant No. DE-SC0012335 (N.M., L.M.). TEM work was supported by the U.S. Department of Energy, Office of Basic Energy Science, Division of Materials Science and Engineering, under Contract No. DE-SC0012704 (L.W., Y.Z.). MEA experiment was the U.S. DOE Energy Efficiency and Renewable Energy, Hydrogen and Fuel Cell Technologies Office, under Contract No. DE-EE0008076 and DE-EE0008417 (Q.Z., C.Z., J.X.). S.T. and E.H. are supported by the Assistant Secretary for Energy Efficiency and Renewable Energy, Vehicle Technology Office of the U.S. DOE through the Advanced Battery Materials Research (BMR) Program under contract no. DE-SC0012704. The authors would like to thank Dr. Elspeth McSweeney from BNL for the valuable discussion. The authors would like to thank Dr. Deborah J. Myers and Dr. Nancy Kariuki from Argonne National Laboratory for ICP-MS measurements.

## Author contributions

K.S. and X.Z. conceived the idea and directed the project. X.Z. synthesized the materials and performed the characterization (STEM/EDS/XRD) and electrochemical experiments. X.Z., N.M., X.C., and L.M. performed the in situ XAS and in situ XRD measurements and data analysis. L.W. and Y.Z. performed the atomic resolution STEM measurements and analysis. Q.Z., C.L., and J.X. performed the MEA fabrication and measurements.

S.T. and E.H. performed the Rietveld refinement analysis. H.C. performed the DFT calculations and machine learning. X.Z. and K.S. wrote the paper with the help of all authors.

## Competing interests

The authors declare no competing interests.
