## [Transparent Peer Review file · Nature Communications]

Sub-angstrom Strain in High-entropy Intermetallic Boosts the Oxygen Reduction Reaction in Fuel Cell Cathodes

Corresponding Author: Dr Kotaro Sasaki

Version 0:

Reviewer comments:

Reviewer #1

(Remarks to the Author)

In the manuscript the synthesis of high entropy intermetallics (HEIs) and nitrogen-doped HEIs as novel ORR catalysts is reported. The synthesis approach has been previously established for N-doped PtNi intermetallics and high entropy alloys. The HEI formation mechanism is followed by in-situ X-ray diffraction. The prepared HEI catalysts show a high oxygen reduction performance and a good stability in both electrochemical measurements with rotating disk electrodes and in a full MEA, outperforming some known high-performance bi- or tri-metallic catalysts. The stability of the catalysts is related to structural features based on DFT simulations.

While the topic of HEI as stable ORR catalysts is of wide interest in the electrocatalysis community, the manuscript is lacking references to relevant literature on high entropy materials as ORR catalysts. The performance of the catalysts is sadly not compared to any other HEI ORR catalysts, where there are catalysts of very high activity such as *J. Am. Chem. Soc.* 2024, 146, 1, 1174–1184, or also *J. Am. Chem. Soc.* 2023, 145, 20, 11140–11150 which the authors cite in the manuscript. Comparison to reported HEI catalysts is crucial, when claiming superior performance. While the authors describe that the materials are superior in stability to PtNi based on the DFT calculations, they do not compare them to their own PtNi experiments. When comparing to the published results (*ACS Catalysis* 10, 10637-10645) with similar catalyst loading (0.1 mg/cm²) it seems that the PtNi outperforms the HEI in both activity and stability. This is an important comparison which is lacking from the manuscript. As the known state of the art ORR catalyst is PtCo, the results should be compared to both PtNi and PtCo prepared by the same synthesis method. Only in this way, a beneficial effect of the high entropy material could be shown. Importantly the materials after ORR also need to be analyzed. In the manuscript, TEM pictures are shown after catalysis. These are missing analysis and discussion. It would be interesting to determine the particle size distribution of the HEI catalysts before and after catalysis and compare to Pt/C and PtNi/C to give more insights into the stability and degradation mechanism, which is now purely based on theoretical simulations. This should be accompanied with a determination of Pt mass or surface area after testing.

For a rigorous determination of the ORR catalyst performance, the Pt surface area should be determined electrochemically by CO stripping and Hupd as it is common practice in the field. While the authors report an ECSA for the catalysts, the determination of the ECSA is missing in the methods. The authors use the Pt mass to estimate the ORR performance and therefore exact composition analysis, such as ICP-MS (with error estimation) of all catalysts should be added. The authors also hypothesize that the transition metals dissolve in the catalysis, which is indeed expected. The dissolution should be determined, e.g. by measurements of the composition after testing. The structural analysis based on the high resolution TEM reports only individual particles – STEM EDS analysis is only shown on a single particle. As the catalysis is however occurring on macroscopic samples, the authors should show more rigorously that the chosen individual particles are representative of the whole sample. In Figure S2 e a white box is on top of the axis description.

The structural analysis of the catalysts includes both in situ XRD and operando XAS measurements. The data collected from these advanced characterization methods lack however analysis. Instead, the authors discuss the strain and defects of the materials, based on the atomic simulation. This should be accompanied with Rietveld analysis of the diffraction patterns. Similarly, the authors hypothesize that the particles form from a Pt core, but they do not analyze the lattice parameter of the in situ XRD. The formation of high entropy materials from Pt cores agrees with literature, which the authors do not refer to, even their own work on HEAs recently published (*J. Am. Chem. Soc.* 2024, 146, 5, 3010–3022). The comparison of the HEIs to the HEAs is also missing from the manuscript.

For the operando XAS, the authors remark that no EXAFS fitting was done due to the complexity of the material. While it is true that these are structurally very complex, the scattering of the non-noble metals should be very similar and it would be possible to devise a simple model resembling of a Pt-TM intermetallic. EXAFS fitting would be important as the Pt-Pt

distance is used to argue that there are differences in strain. Also, it is important to report the electrochemical data of the operando measurements as very different electrochemical cells are used.

The electrochemical data reported are averages of measurements instead of actual electrochemical measurements. While it is good practice to repeat the measurements, these averages should be reported with the error bars from the averaged actual data. The method section on the electrochemical measurements is also not clear, it is missing information of any catalyst pre-conditioning and for a sweep rate is given for a potential step experiment.

Reviewer #2

(Remarks to the Author)

This work is based on a number of advanced characterizations of the catalysts, as well as excellent real-world performance in fuel cells, but there are still a number of issues that need to be resolved seriously before publication.

1. To the best of our knowledge, the concept of high entropy has been reported many times in the field of ORR catalysis, both for alloys and intermetallic compounds, and the innovation of this work is slightly insufficient.
2. As the concept of "sub-angstrom strain" has been mentioned several times in the paper, in addition to XRD, but lacks some more intuitive means of characterization.
3. As mentioned in the paper "optimize the electronic structure of HEI catalysts to enhance catalytic activity but also effectively prevent the TM atoms from dissolving from the core during the ORR process", but there is no direct experimental evidence to prove the loss of metal elements in the catalyst during the reaction process, please carry out the characterization to prove it.
4. Throughout the text, the metal loading and various metal contents of HEI/KB and N-HEI/KB catalysts have only been quantified by SEM-EDS, which in my opinion is inaccurate, please provide ICP data.
5. As mentioned "the increased activity in the early cycles (10k and 20k in Fig. S7) is likely attributed to an increase in number of Pt active sites on the catalyst surface, due to the dissolution of the TM atoms from the surface of N-HEI/KB NPs during the ORR process" in the article, does this indicate in another way that not all metal nanoparticles are intermetallic compounds that form a Pt shell structure during synthesis, and that dissolving the TM shells in an acidic solution exposes the Pt active site?
6. From Fig. S5e, we can see that the intensity of the Pt white line at L-edge of the N-HEI/KB sample is higher than that of the HEI/KB sample, which also suggests that the Pt valence of the N-HEI/KB sample is higher, please explain.
7. In the MEA section, a comparison of the performance of MEAs with different Pt loadings on the cathode shows that the performance of MEA with 0.2 mgPt cm⁻² is worse than the lower loading. However, 0.2 mgPt cm⁻² is actually not a very high loading level, which generally does not cause the problem of poor oxygen transport, and in most of the literature comparisons of low loadings, the performance of the fuel cell improves significantly with increasing Pt loading in the low loadings, please explain.
8. In Fig. S7b, the ECSA of the N-HEI/KB sample decreases with increasing number of ADT cycles, but its ORR performance does not decrease, please explain.

Reviewer #3

(Remarks to the Author)

The manuscript addresses the strain effect in high-entropy intermetallic (HEI) catalysts for oxygen reduction reaction (ORR), which is an underexplored yet important topic. While the combination of atomic-scale characterization and theoretical studies is commendable, the novelty of this work could be better highlighted by explicitly contrasting it with prior studies. It is recommended to include a direct comparison with other high-entropy materials or conventional intermetallic catalysts to clarify the unique contributions of this study. Several major issues:

1. The term "high-entropy effect" is mentioned multiple times but lacks a detailed explanation. The manuscript does not sufficiently elucidate how this effect specifically contributes to the catalyst's performance or structural stability.
2. Does the high-entropy effect play a primary role in corrosion resistance?
3. Are there variations in the contributions of the "pinning effect" and sub-angstrom strain under different cycling conditions?
4. For the quantification of sub-angstrom strain, can additional experimental or theoretical data be provided to further validate the findings?
5. Regarding the stability of the catalyst under MEA testing, a comparative analysis with conventional catalysts over extended cycles would enhance the credibility of the results.

Version 1:

Reviewer comments:

Reviewer #1

(Remarks to the Author)

While the authors have responded to many of the concerns raised in the 1st review, some points still remain open:

The authors have included results obtained from Rietveld refinements of the HEI formation, but not of the N-HEI sample.

Unfortunately, the Rietveld refinements are not shown, also the refined parameters and a goodness of fit is not reported. This must be included as without these data it is not possible to judge whether the model actually fits the data. And it is unclear why a two-phase refinement was necessary from 200-400 degrees C.

The diffraction pattern in Figure 8c does not match the ones from the in-situ experiments. In Figure 8c a clear shoulder is visible which indicates a phase separation, which is most pronounced for the N-HEI samples. Can the authors comment on that?

Also better quality diffraction data as well as Rietveld refinement of these samples (which I assume are the ones actually used for the catalysis) are still missing. Here still the "strain" is shown by drawing lines into a very broad and unsymmetric diffraction peak. This is not sufficient.

The EXAFS fitting shows that the Pt-TM distance is virtually the same, while the Pt-Pt distance varies for N-HEI and HEI. However, the HAADF analysis shows that the Pt-TM distance varies. Can the authors comment on this discrepancy. What is the uncertainty of the HAADF TM position fitting?

In the ECSA determination, CO stripping should be performed and compared to the Hupd, as the determination of surface area is challenging for the multi-component alloys and should not rely on a single method.

While the authors have added ICP-MS data which were requested, the manuscript still relies on STEM-EDS for quantification of the elements. STEM-EDS is a technique which is not well suited for this purpose and needs to be done with utmost care, see e.g. <https://www.sciencedirect.com/science/article/pii/S0968432818301021#bib0190>

Please comment on how quantitative information was obtained from the microscopy and add information on the errors of this estimation.

Reviewer #2

(Remarks to the Author)

The author has effectively addressed my concerns regarding the manuscript.

Reviewer #3

(Remarks to the Author)

The authors have made commendable efforts to address most of the concerns raised in the previous review, and their responses have largely resolved most doubts.

However, I believe the concept of high-entropy and its core effects could still be further elaborated and introduced in greater detail. Given the widespread application of high-entropy strategies across various materials, a more in-depth discussion would enhance the manuscript's scientific rigor and contextual depth. For instance, the authors may consider incorporating insights from studies such as *Energy & Environmental Science* (2021, 14(5): 2883-2905) and *Energy & Environmental Science* (2025, 18: 19-52). Additionally, regarding the calculation of high-entropy sites, it appears that the possibility of multiple-site configurations has been overlooked. Further elaboration and clarification on this aspect would strengthen the discussion.

Version 2:

Reviewer comments:

Reviewer #1

(Remarks to the Author)

The authors adequately addressed all comments raised in the review.

Response to the reviewers' comments

referees,

We sincerely appreciate the reviewers for making constructive comments and valued suggestions, which greatly help us to improve the quality of our manuscript. In the following, we list the point-by-point response to the reviewers' comments. In this Response Letter, the replies are in blue text and the figures are named as Figure R*. All changes have been highlighted by yellow color in the revised Manuscript and Supplementary Information.

REVIEWER COMMENTS

Reviewer #1 (Remarks to the Author):

In the manuscript the synthesis of high entropy intermetallics (HEIs) and nitrogen-doped HEIs as novel ORR catalysts is reported. The synthesis approach has been previously established for N-doped PtNi intermetallics and high entropy alloys. The HEI formation mechanism is followed by in-situ X-ray diffraction. The prepared HEI catalysts show a high oxygen reduction performance and a good stability in both electrochemical measurements with rotating disk electrodes and in a full MEA, outperforming some known high-performance bi- or tri-metallic catalysts. The stability of the catalysts is related to structural features based on DFT simulations.

1. While the topic of HEI as stable ORR catalysts is of wide interest in the electrocatalysis community, the manuscript is lacking references to relevant literature on high entropy materials as ORR catalysts. The performance of the catalysts is sadly not compared to any other HEI ORR catalysts, where there are catalysts of very high activity such as J. Am. Chem. Soc. 2024, 146, 1, 1174–1184, or also J. Am. Chem. Soc. 2023, 145, 20, 11140–11150 which the authors cite in the manuscript. Comparison to reported HEI catalysts is crucial, when claiming superior performance.

[Response]: Thank you for your suggestion. We compared the ORR performance of N-HEI/KB with other HEI and HEA catalysts. As shown in Table R1, the N-HEI/KB catalyst exhibits a high mass activity (MA) of $1.45 \text{ A mg}_{\text{Pt}}^{-1}$ and shows only 17.9% MA loss along with a 6 mV decrease in half-wave potential after 30,000 ADT cycles. These

results surpass the ORR performance of most Pt-based catalysts reported in recent literature, confirming the superior ORR activity and stability of N-HEI/KB catalysts. Table R1 has been added to the revised Supplementary Information as new Supplementary Table 5. The corresponding description can be found on page 9 in the revised Manuscript.

Table R1. Comparison of the electrocatalytic performance of the advanced catalysts reported in recent literatures for oxygen reduction reaction in 0.1 M HClO₄ solutions.

Catalysts	MA @ 0.9 V (A mg _{PGM} ⁻¹)	MA loss (ADT cycles)	Decay of E _{1/2} (mV)	Reference
N-HEI/KB	1.45	17.9% (30k) 32.4% (90k)	5 (30k) 10 (90k)	This work
PtIrFeCoCu-HEI/C	1.29	----	9 (60k)	Ref ¹
PtFeCoNiCuZn-HEI	2.403	5.9% (10k)	~1 (10k)	Ref ²
PtPdCoFeNi HEA/C	1.17	23.6% (50k)	6 (50k)	Ref ³
PtNiFeCuCoZn/PC	0.5	2% (5k)	1 (5k)	Ref ⁴
Pt(FeCoNiCuZn) ₃ /C	0.7	2.9% (30k)	~1 (30k)	Ref ⁵
Pt ₄ FeCoCuNi	3.78	26% (30k)	7 (30k)	Ref ⁶
N-Pt/HEA/C	1.34	20.9% (30k)	8 (30k)	Ref ⁷
L1 ₀ -Pt ₂ CuGa/C	1.39	18.7% (10k)	8 (30k)	Ref ⁸
PtCuCo@Co-N-C	1.14	----	19 (40k)	Ref ⁹
O-Fe ₃ Pt/Ti _{0.5} Cr _{0.5} N	0.673	9.7% (5k)	6 (5k)	Ref ¹⁰
Gd-O-Pt ₃ Ni	1.54	27.9% (70k)	8 (70k)	Ref ¹¹
FePt@PtBi	0.96	18% (30k)	----	Ref ¹²

References:

- 1 Feng, G. *et al.* Engineering structurally ordered high-entropy intermetallic nanoparticles with high-activity facets for oxygen reduction in practical fuel cells. *J. Am. Chem. Soc.* **145**, 11140-11150 (2023).
- 2 Chen, T. *et al.* An ultrasmall ordered high-entropy intermetallic with multiple active sites for the oxygen reduction reaction. *J. Am. Chem. Soc.* **146**, 1174-1184 (2024).
- 3 Yu, Y. *et al.* High-entropy alloy nanoparticles as a promising electrocatalyst to enhance activity and durability for oxygen reduction. *Nano Res.* **15**, 7868-7876 (2022).
- 4 Zhang, W., Feng, X., Mao, Z. X., Li, J. & Wei, Z. Stably immobilizing sub-3 nm high-entropy Pt alloy nanocrystals in porous carbon as durable oxygen reduction electrocatalyst. *Adv. Funct. Mater.* **32** (2022).
- 5 Zhang, Q. *et al.* High-entropy L₁₂-Pt(FeCoNiCuZn)₃ intermetallics for ultrastable oxygen reduction reaction. *J. Energy Chem.* **86**, 158-166 (2023).
- 6 Wang, Y. *et al.* Ordering-dependent hydrogen evolution and oxygen reduction electrocatalysis of high-entropy intermetallic Pt₄FeCoCuNi. *Adv. Mater.* e2302067 (2023).
- 7 Zhao, X. *et al.* Multiple Metal-nitrogen bonds synergistically boosting the activity and durability of high-entropy alloy electrocatalysts. *J. Am. Chem. Soc.* **146**, 3010-3022 (2024).
- 8 Liu, X. *et al.* Inducing covalent atomic interaction in intermetallic Pt alloy nanocatalysts for high-performance fuel cells. *Angew. Chem. Int. Ed.* **62**, e202302134 (2023).
- 9 Huang, L. *et al.* Boosting oxygen reduction via integrated construction and synergistic catalysis of porous platinum alloy and defective graphitic carbon. *Angew. Chem. Int. Ed.* **60**, 25530-25537 (2021).
- 10 Liu, Q. *et al.* Structurally ordered Fe₃Pt nanoparticles on robust nitride support as a high performance catalyst for the oxygen reduction reaction. *Adv. Energy Mater.* **9** (2019).
- 11 Yang, L. *et al.* Rare earth evoked subsurface oxygen species in platinum alloy catalysts enable durable fuel cells. *Angew. Chem. Int. Ed.* **63**, e202315119 (2024).
- 12 Guan, J. *et al.* Intermetallic FePt@PtBi core-shell nanoparticles for oxygen reduction electrocatalysis. *Angew. Chem. Int. Ed.* **60**, 21899-21904 (2021).

2. While the authors describe that the materials are superior in stability to PtNi based on the DFT calculations, they do not compare them to their own PtNi experiments. When comparing to the published results (ACS Catalysis 10, 10637-10645) with similar catalyst loading (0.1 mg/cm²) it seems that the PtNi outperforms the HEI in both activity and stability. This is an important comparison which is lacking from the manuscript. As the known state of the art ORR catalyst is PtCo, the results should be compared to both PtNi and PtCo prepared by the same synthesis method. Only in this way, a beneficial effect of the high entropy material could be shown.

[Response]: We appreciate the reviewer's insightful comments regarding the comparison of our HEI to PtNi and PtCo catalysts. First, we would like to clarify that the MEA performance of PtNi in the published paper (*ACS Catalysis*, 2020, 10, 10637-10645) was evaluated under light-duty vehicle (LDV) conditions, while the N-HEI was tested under heavy-duty truck (HDV) conditions. As such, direct comparison is challenging due to the different operating conditions. Nevertheless, we agree that a comparison with PtNi and PtCo is crucial and have addressed this in our revised manuscript.

We synthesized nitrogen-doped intermetallic PtNi and PtCo catalysts supported on Ketjenblack (PtNiN/KB and PtCoN/KB) by annealing in a flowing ammonia gas. As shown in Figure R1a-b, the X-ray diffraction (XRD) patterns indicate the generation of the ordered intermetallic phase of PtNi and PtCo, with additional (001) and (110) peaks at around 24.7 and 33.4°, respectively. Additionally, we evaluated the ORR performance of these catalysts using rotating disk electrode (RDE) measurements, as shown in Figure R1, and compared their performance to that of the HEI catalysts. As depicted in Figure R2, the N-HEI catalyst shows superior ORR activity and stability compared to PtNiN/KB and PtCoN/KB. Although the mass activity of HEI/KB is slightly lower than that of PtNiN/KB, the durability of HEI/KB is significantly higher, suggesting the enhanced structural stability of HEI materials.

These results have been added in the revised Supplementary Information (Supplementary Fig. 8 and Table 4) and Manuscript (Fig. 2c). The corresponding description can be found on page 9 in the revised Manuscript.

Figure R1. XRD and ORR performance of PtNiN/KB and PtCoN/KB. a. XRD patterns of PtNiN/KB. b. XRD patterns of PtCoN/KB (Cu $K\alpha$, wavelength 1.5406 Å). c-d. ORR polarization curves and CV curves of PtNiN/KB at BOL and ADT 30,000 cycles. e-f. ORR polarization curves and CV curves of PtCoN/KB at BOL and ADT 30,000 cycles.

Figure R2. Mass activity of N-HEI/KB, HEI/KB, PtNiN/KB, PtCoN/KB, and commercial Pt/C catalysts at BOL and 30k cycles.

3. Importantly the materials after ORR also need to be analyzed. In the manuscript, TEM pictures are shown after catalysis. These are missing analysis and discussion. It would be interesting to determine the particle size distribution of the HEI catalysts before and after catalysis and compare to Pt/C and PtNi/C to give more insights into the stability and degradation mechanism, which is now purely based on theoretical simulations. This should be accompanied with a determination of Pt mass or surface area after testing.

[Response]: Thank you for the valuable suggestions. We added the analysis and discussions for all catalysts (N-HEI/KB, HEI/KB, PtNiN/KB, PtCoN/KB and commercial Pt/C) after ADT cycles.

As shown in Figure R3, both N-HEI/KB and HEI/KB maintain their morphology after 30,000 ADT cycles, exhibiting no obvious nanoparticle loss or agglomeration. This highlights the exceptional structural stability of the HEI catalysts. In contrast, STEM images of PtNiN/KB, PtCoN/KB, and the commercial Pt/C reveal clear particle agglomeration and an increase in particle size (Figure R4). Despite some inevitable dissolution of transition metal elements, EDS analysis shows that all elements in N-HEI/KB and HEI/KB are still uniformly distributed in the nanoparticle cores (Figure R3). In comparison, PtNiN/KB and PtCoN/KB exhibit greater transition elements dissolution (Figure R4 and Table R2), further emphasizing the superior compositional stability of the HEI structures.

These results have been added in the revised Supplementary Information

(Supplementary Figs. 9-10 and Table 6). The corresponding description was added to the revised Manuscript (Pages 9-10).

Figure R3. Structure and morphology of N-HEI/KB and HEI/KB catalyst after 30k ADT cycles in RDE testing. a. STEM-HAADF images and the EDS mappings of the N-HEI/KB catalyst after ADT 30k cycles. b. STEM-HAADF images and the EDS elemental mappings of the HEI/KB catalyst after ADT 30k cycles.

Figure R4. Structure and morphology of PtNiN/KB, PtCoN/KB, and Pt/C catalysts before and after ADT cycles in RDE testing. a-c. STEM-HAADF images and the EDS mappings of PtNiN/KB, PtCoN/KB, and commercial Pt/C at BOL. d-f. STEM-HAADF images and the EDS mappings of PtNiN/KB, PtCoN/KB, and commercial Pt/C after 30k ADT cycles.

Table R2. Element loading on the carbon support for different catalysts at BOL and after 30k cycles, based on the average of 10 STEM-EDS results.

Samples	BOL		30k cycles		Element loss	
	Pt (wt.%)	M* (wt.%)	Pt (wt.%)	M* (wt.%)	Pt (%)	M* (%)
N-HEI/KB	22.4	6.7	21.8	5.8	-2.7%	-13.4%
HEI/KB	22.1	6.2	21.4	4.7	-3.2%	-24.2%
PtNiN/KB	22.6	6.5	20.3	3.6	-10.2%	-44.6%
PtCoN/KB	23.5	5.8	21.2	1.8	-9.8%	-69.0%
Pt/C	46.1	----	34.7	----	-24.7%	----

*M refers to the transition metals in different Pt-based catalysts. Specifically, M represents the total loading of transition metals (Co/Fe/Ni/Cu) in N-HEI/KB and HEI/KB.

4. For a rigorous determination of the ORR catalyst performance, the Pt surface area should be determined electrochemically by CO stripping and Hupd as it is common practice in the field. While the authors report an ECSA for the catalysts, the determination of the ECSA is missing in the methods. The authors use the Pt mass to estimate the ORR performance and therefore exact composition analysis, such as ICP-MS (with error estimation) of all catalysts should be added. The authors also hypothesize that the transition metals dissolve in the catalysis, which is indeed expected. The dissolution should be determined, e.g. by measurements of the composition after testing.

[Response]: We sincerely appreciate these insightful comments. The electrochemical surface area (ECSA) was determined using the underpotentially deposited hydrogen (Hupd) method, which was calculated from the averaged charge of hydrogen adsorption and desorption peaks. We have now included the ECSA determination in the "Electrochemical Measurements" section of the revised manuscript (Page 19).

As suggested, we have added the ICP-MS data of N-HEI/KB and HEI/KB. As shown

in Table R4 below (Reviewer 2, comment 4), the Pt content in the HEI/KB and N-HEI/KB was determined to be 23.9 wt.% and 23.6 wt.% by inductively coupled plasma–mass spectroscopy (ICP-MS), respectively. These values are similar to those obtained from the EDS analysis (Table R2). In this work, the Pt loading of samples is about 12 $\mu\text{g cm}^{-2}$ (based on 23 wt.% Pt content), which was used to calculate mass activity. This information has been added to the revised Manuscript (Page 19). The ICP-MS results are now included in the revised Supplementary Information (Supplementary Table 2).

We have also provided the composition of Pt and the total transition metals for all catalysts before and after 30k ADT cycles, as shown in Table R2 above. Compared to PtNiN/KB and PtCoN/KB, both N-HEI/KB and HEI/KB exhibited less Pt mass loss and lower transition metal dissolution, further confirming the superior structural and compositional stability of the HEI structures. It is important to note that the slightly lower element content values observed in the EDS measurements may be due to the lacey carbon grid used for STEM analysis. However, this does not affect the overall trend.

5. The structural analysis based on the high resolution TEM reports only individual particles – STEM EDS analysis is only shown on a single particle. As the catalysis is however occurring on macroscopic samples, the authors should show more rigorously that the chosen individual particles are representative of the whole sample. In Figure S2 e a white box is on top of the axis description.

[Response]: We thank this reviewer’s careful reading of our manuscript. In response, we have added additional STEM-EDS and high-resolution STEM analyses of multiple nanoparticles in the N-HEI and HEI catalysts. The STEM-EDS results demonstrate that all elements are uniformly distributed within the N-HEI and HEI nanoparticles (Figure R5e and Figure R6e). High-resolution STEM images revealed a clear $L1_0$ -ordered structure in multiple N-HEI and HEI nanoparticles (Figure R7a, 7c), confirming that the selected individual particles are representative. These findings have been added to the revised Supplementary Information (Supplementary Figs. 2-4) and the revised Manuscript (Pages 5-6).

Additionally, the white box in Fig. S2e (now Supplementary Fig. 2f in the revised Supplementary Information) has been removed.

Figure R5. Morphology of HEI/KB catalyst. a. TEM image of HEI/KB catalyst. b. STEM-HAADF image of HEI/KB catalyst. c. The particle size distribution of the HEI/KB catalyst. d-e. STEM-HAADF image and the corresponding EDS elemental mappings of single and multiple HEI nanoparticles. f. STEM-EDS line scan profiles of HEI/KB along the yellow single line in d.

Figure R6. Morphology of N-HEI/KB catalyst. a. TEM image of the HEI/KB catalyst. b. STEM-HAADF image of N-HEI/KB catalyst. c. The particle size distribution of the N-HEI/KB catalyst. d-e. STEM-HAADF image and the corresponding EDS elemental mappings of single and multiple N-HEI nanoparticles. f. STEM-EDS line scan profiles of N-HEI/KB along the yellow single line in d.

Figure R7. Atomic resolution STEM-HAADF images of a-b. N-HEI/KB and c-d. HEI/KB. The ordered phase in a and c are indicated by yellow dotted circles. The arrows show the displacement of TM atoms from the center of four Pt atoms in b and d. The positions of Pt and TM are refined based on the second polynomial function. Scale bar 2 nm for b and d.

6. The structural analysis of the catalysts includes both in situ XRD and operando XAS measurements. The data collected from these advanced characterization methods lack however analysis. Instead, the authors discuss the strain and defects of the materials, based on the atomic simulation. This should be accompanied with Rietveld analysis of the diffraction patterns. Similarly, the authors hypothesize that the particles form from a Pt core, but they do not analyze the lattice parameter of the in situ XRD. The formation of high entropy materials from Pt cores agrees with literature, which the authors do not refer to, even their own work on HEAs recently published (J. Am. Chem. Soc. 2024, 146, 5, 3010–3022). The comparison of the HEIs to the HEAs is also missing from the manuscript.

[Response]: This is a good suggestion. We performed the Rietveld refinement analysis for *in-situ* XRD data of HEI/KB and compared the lattice constant of the HEI and HEA.

As shown in Figure R8a, we termed the temperature region up to 220 °C as stage I (Pt reduction), 220-740 °C as stage II (HEA generation), and above 740 °C as stage III (HEI generation). The Rietveld refinement analysis show that the lattice constant of the sample at 150 °C is about 3.891 Å, which is close to that of pure Pt metal (3.920 Å), but much larger than those of Co (3.537 Å), Ni (3.524 Å), Fe (2.866 Å), and Cu (3.615 Å). This value shows negligible change until 220 °C, pointing to the formation of Pt-rich phases in the NPs during stage I. The Pt-rich NPs act as seeds for the formation of high-entropy materials and the observation is in accord with our previous study (*J. Am. Chem. Soc.* 2024, 146, 5, 3010–3022). At around 220 °C, the lattice constant from a new (111) peak at 16.81° is about 3.673 Å, significantly smaller than that of Pt, suggesting the alloying of 3d TMs with the Pt-rich NPs. During stage II, the lattice constant of Pt-rich NPs declines with rising temperatures, while the lattice constant of alloys increases considerably, suggesting a progressive increase in the level of alloying as the TM atoms steadily integrate into the face-centered cubic (fcc) lattice of Pt. During stage III, the lattice constant of sample increases with rising temperature, further supporting the transition from a disordered to an ordered phase.

The Rietveld refinement analysis have been added to the revised Supplementary Information (Supplementary Fig. 1a). The corresponding description was added in the revised Manuscript (Page 4).

Figure R8. a. Rietveld refinement analysis showing changes in lattice constant of the HEI nanoparticles as a function of annealing temperatures. b. In situ synchrotron XRD patterns of HEI/KB holding at 850 °C in 10% H₂/He gas stream (wavelength 0.6199 Å). c. XRD patterns of N-HEI/KB and HEI/KB catalysts (Cu K α , wavelength 1.5406 Å).

7. For the operando XAS, the authors remark that no EXAFS fitting was done due to the complexity of the material. While it is true that these are structurally very complex, the scattering of the non-noble metals should be very similar and it would be possible to devise a simple model resembling of a Pt-TM intermetallic. EXAFS fitting would be important as the Pt-Pt distance is used to argue that there are differences in strain. Also, it is important to report the electrochemical data of the operando measurements as very different electrochemical cells are used.

[Response]: We thank the reviewer's instructive comment. We performed the fitting of Pt L₃ spectra of the N-HEI and HEI catalysts at 0.42 V in 0.1 M HClO₄, by lumping Co, Ni, Fe, and Cu as one transition metal (TM). We used Ni for representing TM of Pt-TM fitting in FEFF calculation in Artemis. The fitting results are shown in Figure R9 and

listed in Table R3. The Pt-Pt distance of the N-NEI/KB catalyst is 2.7104 Å, which is longer than that of the HEI/KB catalyst (2.6983Å). We consider that the longer Pt-Pt distance in the N-NEI/KB catalyst could be caused by the enhanced strain in the TM layers due to doped N atoms.

The fitting results have been added to the revised Supplementary Information (Supplementary Fig. 13 and Table 7). The corresponding description was added in the revised Manuscript (Page 10).

Figure R9. Pt L_3 -edge FT-EXAFS spectra of (a) N-HEI/KB and (b) HEI/KB catalysts at 0.42 V with a first-shell fit together with Pt-Pt and Pt-TM contributions. The *in-situ* data were collected in a 0.1 M $HClO_4$ electrolyte using an electrochemical cell (see below Figure R10)

Table R3. Fitting results of bonding distances (R) and coordination numbers (CN) of Pt-Pt and Pt-TM pairs of the N-HEI/KB and HEI/KB catalysts.

	R(Pt-Pt) (Å)	R(Pt-TM) (Å)	CN(Pt-Pt)	CN(Pt-TM)
N-HEI/KB	2.7104 (± 0.0048)	2.6342 (± 0.0048)	6.9 (± 0.8)	3.2 (± 0.6)
HEI/KB	2.6983 (± 0.0030)	2.6315 (± 0.0051)	5.0 (± 0.4)	3.0 (± 0.4)

For the comment to report the electrochemical data of the operando measurements. As the reviewer pointed out, we used the different electrochemical cells for *in-situ* XAS measurement, which resulted in different ORR performance. The key distinction is that *in-situ* XAS measurements focus on evaluating the structural changes under the same voltage in a half-cell configuration, rather than replicating the ORR performance under

RDE conditions. To acknowledge the reviewer's observation and provide further clarity, we have included the *in-situ* cell and corresponding electrochemical data for N-HEI/KB in Figure R10. Additionally, our previously published works (*J. Am. Chem. Soc.* 2024, 146, 3010–3022; *J. Am. Chem. Soc.* 2023, 145, 19076–19085; *ACS Catal.* 2020, 10, 10637–10645) using the same cell design can further support the validity of the *in-situ* measurement setup.

Figure R10. a. Schematic diagrams of *in-situ* electrochemical cell. b. *in-situ* electrochemical data of N-HEI/KB under different potentials.

8. The electrochemical data reported are averages of measurements instead of actual electrochemical measurements. While it is good practice to repeat the measurements, these averages should be reported with the error bars from the averaged actual data. The method section on the electrochemical measurements is also not clear, it is missing information of any catalyst pre-conditioning and for a sweep rate is given for a potential step experiment.

[Response]: Regarding the electrochemical data, we understand the concern and agree that reporting error bars is important for clarity and reproducibility. In this work, we repeated the electrochemical measurements more than three times. We selected three ORR curves to calculate the error bars for the mass activity, and the electrochemical curves (LSV and CV) presented in this work correspond to the median curves. As

shown in Figure R2, we included the error bars for the mass activity of all catalysts at BOL and after 30k cycles. And the corresponding description was added in the revised Manuscript (Page 20).

We apologize for the lack of clarity regarding the electrochemical measurements. We have now revised the "Electrochemical Measurements" section to include details on the catalyst pre-conditioning procedure and the sweep rate used in the potential step experiments. The revised section is as follows:

“The cyclic voltammetry (CV) characterization of the catalysts was carried out in an Ar-purged 0.1 M HClO₄ aqueous solution. CV was first performed for 20 scans at a scan rate of 100 mV s⁻¹ to clean and stabilize the catalysts, then followed by measurements at a scan rate of 20 mV s⁻¹. The electrochemical surface area (ECSA) was determined using the underpotentially deposited hydrogen (Hupd) method, which was calculated from average hydrogen adsorption and desorption peaks. The linear scanning voltammetry (LSV) for the ORR was obtained in an O₂-saturated 0.1 M HClO₄ solution. LSV was first performed for 10 scans at a sweep rate of 100 mV s⁻¹ to stabilize the curves, then was measured by scanning the potential from 0 to 1.1 V with iR correction (scan rate: 10 mV s⁻¹; rotation rate: 1600 rpm).” (Page 19 in Manuscript)

Reviewer #2 (Remarks to the Author):

This work is based on a number of advanced characterizations of the catalysts, as well as excellent real-world performance in fuel cells, but there are still a number of issues that need to be resolved seriously before publication.

1. To the best of our knowledge, the concept of high entropy has been reported many times in the field of ORR catalysis, both for alloys and intermetallic compounds, and the innovation of this work is slightly insufficient.

[Response]: We appreciate the reviewer’s feedback. To address concerns about the innovation, we have made significant revisions to the Introduction in revised Manuscript (Page 3). Specifically, we have clarified the following points:

1. In HEA catalysts, traditional lattice distortions are caused by mismatches of ionic size, mass and valence-electron configuration. While in HEI catalysts, there is not only strain arise from atomic size differences but also the displacement of TM atoms within

the TM layers, known as “sub-angstrom strain”. This occurs because the TM atoms in HEI catalysts occupy the sites in each sublattice with local clustering/segregation, driven by the stabilizing effect of Pt layers. Fundamentally, the key distinction between strain in HEA and HEI catalysts lies in the dominant mechanism: the local segregation-induced sub-angstrom strain in HEI catalysts and the strain induced by atomic size differences in HEA catalysts. So far, a comprehensive understanding of the relationship between structure and electrocatalytic performance of HEI catalysts has remained elusive. It is critical to explore the catalytic mechanisms of HEI catalysts.

2. Previous research on Pt-based intermetallic catalysts has primarily focused on the strain effects on Pt atoms, with little attention given to the strain interactions (e.g., displacement of TM atoms) between different transition metals within the TM layers. This oversight highlights a gap in understanding the full potential of HEI materials.

3. Notably, this study represents the first instance of revealing the origin of the sub-angstrom strain in HEI catalysts by combining atomic-scale characterization and theoretical studies. The effect of sub-angstrom strain is further enhanced by N-doping, resulting in excellent ORR performance as demonstrated in the present work.

Based on the above three points, we are confident that the revised manuscript clearly defines the specific scientific advancement and significance of our findings. The work on the N-HEI catalysts for ORR, emphasizing the sub-angstrom strain effect and HDV applications, will resonate with fellow specialists.

2. As the concept of “sub-angstrom strain” has been mentioned several times in the paper, in addition to XRD, but lacks some more intuitive means of characterization.

[Response]: In this study, we have used atomic resolution STEM-HAADF, in which we believe the best characterization tool to provide direct and clear insights into the “sub-angstrom strain” at the atomic scale. Figures R11e-f visualize the discernible atomic column displacements of TMs from the lattice center as indicated by the arrows, where the head and tail of the arrow represent the direction and amplitude (A) of the displacement, respectively. This is also illustrated schematically in Figure R11g. In contrast to conventional discussion on lattice distortions resulting from mismatches in ionic size and mass, these atomic resolution images directly show the sub-angstrom displacements of TMs in HEI structures, in which we define as "sub-angstrom strain". We have clarified this in the revised Manuscript (Page 3).

Additionally, to provide more intuitive evidence of the sub-angstrom strain, we added the atomic resolution STEM-HAADF image of N-doped L1₀-ordered PtNi catalyst (PtNiN/KB) for comparison. As shown in Figure R12, there is basically no discernible amplitude in displacement of TMs in the binary PtNi particle, further supporting the existence of the unique sub-angstrom strain in HEI structures.

Figure R12 has been added to the updated Supplementary Fig. 4 in the revised Supplementary Information. The corresponding description was added in the revised Manuscript (Page 6).

Figure R11. Structure evolution and properties of the N-HEI/KB catalyst. (a-b) In situ synchrotron XRD patterns for the structure evolution of HEI/KB during annealing in 10% H₂/He gas stream (wavelength 0.6199 Å). (c) Schematic for the synthesis of HEI/KB. (d) Representative atomic-resolution STEM-HAADF image of N-HEI particle viewed along [-110] direction. The image is lightly filtered in frequency space by applying a periodic mask to remove noise. The image contrast is approximately proportional to $Z^{1.7}$ along the atomic column, thus the dots with strong and weak image contrast correspond to Pt and TM columns, respectively. (e) Enlarged image from the area marked by red rectangle in (d). The white rectangle marks the lattice of N-HEI structure. (f) STEM-HAADF image from part of a HEI particle. The yellow spheres mark the Pt atoms at the edge. Scale bars are 2 nm for (d) and 0.5 nm for (e, f). (g)

Schematic of the sub-angstrom displacement of TMs. Four lattices are outlined by solid lines with Pt atoms at the apex. The arrows show the displacement of the TM from the center of the lattice (cross of the dashed lines) with their head and tail representing the direction and amplitude (A) of the displacement. (h) Intensity profiles (small circles) from scan lines in (e) and (f) with the same color (blue for N-HEI and red for HEI). The solid lines are refined curves using Gaussian function. The green, pink, and black vertical lines mark the positions of Pt columns, center of the lattice, and TM columns, respectively.

Figure R12. a. Atomic resolution STEM-HAADF image of PtNiN/KB. b. Enlarged image from the area marked by blue rectangle in (a). The arrows show the displacement of the Ni from the center of the lattice with their head and tail representing the direction and amplitude of the displacement. Scale bars are 2 nm for (a) and 0.5 nm for (b).

3. As mentioned in the paper “optimize the electronic structure of HEI catalysts to enhance catalytic activity but also effectively prevent the TM atoms from dissolving from the core during the ORR process”, but there is no direct experimental evidence to prove the loss of metal elements in the catalyst during the reaction process, please carry out the characterization to prove it.

[Response]: To address the concern regarding the loss of transition metal (TM) elements during the ORR process, we have included composition analysis of the N-HEI and HEI catalysts after the ORR measurements. Additionally, we compared the TM element loss in PtNiN/KB and PtCoN/KB during the reaction process.

As shown in the above Table R2 (Reviewer 1, comment 3), the TM element loss of different catalysts after 30k ADT cycles is as follows:

N-HEI/KB (13.4%) < HEI/KB (24.2%) < PtNiN/KB (44.6%) < PtCoN/KB (69.0%)

These results provide experimental evidence that the HEI structure significantly reduces TM dissolution, thereby improving the durability of the catalysts. Table R2 was added in the revised Supplementary Information (Supplementary Table 6).

4. Throughout the text, the metal loading and various metal contents of HEI/KB and N-HEI/KB catalysts have only been quantified by SEM-EDS, which in my opinion is inaccurate, please provide ICP data.

[Response]: Thank you for your suggestion. We have added the ICP data of N-HEI/KB and HEI/KB. As shown in Table R3, the Pt content in the HEI/KB was determined to be 23.9 wt.% by inductively coupled plasma–mass spectroscopy (ICP-MS). The atomic ratio of Pt to total TMs in both HEI/KB and N-HEI/KB is close to 1:1, which aligns with the EDS results (Supplementary Table 1). Table R2 has been added to the revised Supplementary Information (Supplementary Table 2), and the corresponding description has been added into the revised Manuscript (Page 5).

Additionally, we would like to clarify that SEM/STEM-EDS was used to quantify the metal loading and contents based on two key points: 1) SEM/STEM-EDS can analyze a wide range of elements, including light elements like carbon and nitrogen, which is crucial for our N-doped HEI samples; 2) more than 10 regions were selected for SEM/STEM-EDS signal collection for all the samples, and averages were taken. The EDS results match well with the ICP-MS results, further confirming the reliability of the EDS results in this work.

Table R4. The element analysis of HEI/KB and N-HEI/KB by ICP-MS.

Sample	Pt (wt.%)	Co (wt.%)	Ni (wt.%)	Fe (wt.%)	Cu (wt.%)	Atomic ratio
HEI/KB	23.9	1.61	1.72	1.66	1.94	Pt ₅₁ Co ₁₂ Ni ₁₂ Fe ₁₂ Cu ₁₃
N-HEI/KB	23.6	1.64	1.78	1.66	2.02	Pt ₅₀ Co ₁₂ Ni ₁₃ Fe ₁₂ Cu ₁₃

5. As mentioned “the increased activity in the early cycles (10k and 20k in Fig. S7) is likely attributed to an increase in number of Pt active sites on the catalyst surface, due to the dissolution of the TM atoms from the surface of N-HEI/KB NPs during the ORR process” in the article, does this indicate in another way that not all metal nanoparticles are intermetallic compounds that form a Pt shell structure during synthesis, and that dissolving the TM shells in an acidic solution exposes the Pt active site?

[Response]: Yes, this observation indicates that not all metal nanoparticles are intermetallic compounds that form a Pt shell structure during synthesis. To clarify further, we conducted the quantitative analysis of the ordering degree for N-HEI/KB and HEI/KB based on the XRD results (as shown in Figure R8c above). The ordering degrees for N-HEI/KB and HEI/KB are 94.9 % and 96.3 %, respectively, which are calculated by the ratio of the (110) peak area to the combined area of the (111) and (200) peaks, denoted as $S(110)/(S(111) + S(200))$. This ratio is employed as a quantitative measure of the ordering degree, as it exhibits a direct linear correlation with the ordering degree (*Proc. Natl. Acad. Sci. USA 2019, 116, 1974*).

Since the ordering degree is not 100%, some of the Pt shell structure in the HEI catalysts is not perfect, causing the transition metals on the surface to dissolve into an acidic solution during the ORR process. This may expose additional Pt active sites or optimize the surface structure of N-HEI, thereby increasing the ORR activity (*Joule 3, 124–135, 2019*).

The ordering degree results, and the corresponding description have been added in revised Manuscript (Page 5).

6. From Fig. S5e, we can see that the intensity of the Pt white line at L-edge of the N-HEI/KB sample is higher than that of the HEI/KB sample, which also suggests that the Pt valence of the N-HEI/KB sample is higher, please explain.

[Response]: We appreciate the reviewer's observation that the higher intensity of the Pt white line (WL) at the L-edge may indicate a higher Pt valence. However, it is important to note that several factors can influence the WL intensity at the Pt L₃-edge, including the d-band vacancy in Pt, the electron density of states, and lattice distortion, et al. In our N-HEI/KB sample, the formation of Pt-N bonds is challenging due to the absence of the high-pressure and high-temperature conditions (~50 GPa and 2000 K) typically required for their formation (*Science 2006, 311(5765),1275–1278*). As such, it is difficult to conclusively correlate the higher intensity of the Pt white line with an

increased Pt valence when comparing N-HEI/KB and HEI/KB.

In this work, we consider the origin of the higher white (WL) intensity of the N-HEI/KB catalyst as follows: the *ex-situ* hard XAS of K-edges (Supplementary Fig. 5) and soft XAS of L-edges (Supplementary Fig. 6) of TMs of the N-HEI/KB catalyst showed the higher WL intensity and the WL peak shifts to higher energy, respectively, compared to those of the HEI/KB catalyst. The XAS results indicate elevated oxidation states of the TMs in the N-HEI/KB catalyst, due to the formation of multiple TM-N bonds and the resultant electron transfer from TMs to N atoms. This induces an increase in d-band vacancies in the TMs, which in turn enhance electron transfer from Pt to the TMs, according to Hammer and Nørskov's model (*Adv. Catal.* 45, 2-71, 2000). Consequently, the d-band vacancy in Pt increases, which can enhance the WL intensity in Pt L₃ edge (*Res. Chem. Intermed.* 32, 543-559, 2006). Thus, it is envisaged that the higher WL intensity of the N-HEI/KB than the HEI/KB is due to the increase in d-band vacancy of the Pt, which is induced by the increased d-band vacancies of the TMs through the formation of multiple TM-N bonds.

The corresponding description has been added in the revised Manuscript (Page 7).

7. In the MEA section, a comparison of the performance of MEAs with different Pt loadings on the cathode shows that the performance of MEA with 0.2 mg_{Pt} cm⁻² is worse than the lower loading. However, 0.2 mg_{Pt} cm⁻² is actually not a very high loading level, which generally does not cause the problem of poor oxygen transport, and in most of the literature comparisons of low loadings, the performance of the fuel cell improves significantly with increasing Pt loading in the low loadings, please explain.

[Response]: We agree with the reviewer that the MEA performance generally increases with the Pt loading when Pt loading is 0.2 mg_{Pt}/cm² and beyond. However, in this work, we observed that the MEA performance at 0.7V improved while stability slightly decreased as the catalyst Pt loading decreased. This is apparently counterintuitive, and we try to explain such a phenomenon.

Electrode thickness and microstructure significantly influences reaction kinetics and mass transport in fuel cells. While higher Pt loading generally enhances fuel cell performance due to increased catalyst availability, other critical factors, such as catalyst layer morphology, ionomer-to-carbon (I/C) ratio, and oxygen transport resistance, also play a crucial role. In our work, the Pt content in the catalysts is only 23 wt.%, meaning

that at a given Pt loading, the catalyst layer consists of a larger fraction of carbon and ionomer, which can impact both charge transfer resistance, mass transfer resistance and Pt utilization. At lower Pt loadings (such as 0.05 and 0.10 mg_{Pt} cm⁻²), the catalyst is more effectively dispersed, allowing for higher electrochemical surface area (ECSA) and improved H⁺ and oxygen access. This leads to enhanced mass transport properties and superior MEA performance at the beginning of life (BOL), even though stability might decrease over time (*ACS Appl. Mater. Interfaces* 2022, 14, 36731–36740). It has been proved that the total oxygen transport resistance in the electrode comes from both diffusion through the electrode pores (across the thickness of the electrode) and from a resistance which inversely scales rapidly with available platinum surface area (*Journal of The Electrochemical Society* 2012, 159 (12), F831-F840).

For ultra-low catalyst loading cathode, the much thinner catalyst layer would significantly accelerate the O₂ access on the Pt nanoparticles, reduce the O₂ transfer resistance and thus results in a higher oxygen local concentration, leading to high reaction rate. Consequently, in our system, the MEA with 0.2 mg_{Pt} cm⁻² may exhibit greater mass transport limitations due to a thicker catalyst layer and less optimal ionomer distribution, counteracting the expected benefit of higher Pt loading. While some literatures show improved performance with increasing Pt loading at certain levels, this trend is highly dependent on system-specific factors such as Pt particle size, catalyst layer porosity, ionomer content, and electrode structure. These parameters influence oxygen and proton transport pathways, as well as the effective utilization of Pt sites, leading to variations in observed performance trends.

8. In Fig. S7b, the ECSA of the N-HEI/KB sample decreases with increasing number of ADT cycles, but its ORR performance does not decrease, please explain.

[Response]: Thank you for your comment. Here, we would like to clarify two key points:

1. The difference between ECSA and mass activity (MA): **ECSA** refers to the electrochemical surface area, which is an estimate of the available surface area of active sites for catalysis; **MA** refers to the catalytic activity per unit of the catalyst's mass, often expressed in terms of current density. In short, ECSA mainly reflects the *quantity* of active sites, whereas MA gives insight into the *quality* and *efficiency* of those sites. As a result, ECSA and MA may not always exhibit the same trend due to many factors,

such as surface restructuring and improved intrinsic activity.

2. The decrease in ECSA could be attributed to (i) particle migration and coalescence and/or (ii) Ostwald ripening, where smaller nanoparticles dissolve, contributing to particle growth and consequently affecting the overall surface area (*Top. Catal.* 2007, 46, 285-305). However, despite this decrease in ECSA, the MA still improves in the early cycles (10k and 20k). This is likely because the dissolution of TM atoms exposes additional Pt sites on the surface, leading to a structural reconstruction (such as the reduction of low-coordinated sites), which enhances the specific activity (SA). Since $MA = SA \times ECSA$, the increase in SA can compensate for the loss of ECSA, allowing the MA to increase or remain unchanged despite the decrease in ECSA. Thus, the observed stability in ORR performance can be attributed to this enhanced SA, which more than offsets the impact of the decreased ECSA in the early cycles.

Based on the above clarification, we have also revised the sentence for Supplementary Fig. 7b in the updated Manuscript, as follows:

“The increased MA and slightly decreased ECSA in the early cycles (10k and 20k cycles, as shown in Supplementary Fig. 7) are likely attributed to an increase in SA due to increasing Pt active sites, caused by the dissolution of the TM atoms from the surface of N-HEI/KB NPs and the consequent structural reconstruction on the catalyst surface, such as reducing low-coordinated sites.” (Page 9)

Reviewer #3 (Remarks to the Author):

The manuscript addresses the strain effect in high-entropy intermetallic (HEI) catalysts for oxygen reduction reaction (ORR), which is an underexplored yet important topic. While the combination of atomic-scale characterization and theoretical studies is commendable, the novelty of this work could be better highlighted by explicitly contrasting it with prior studies. It is recommended to include a direct comparison with other high-entropy materials or conventional intermetallic catalysts to clarify the unique contributions of this study. Several major issues:

1. The term "high-entropy effect" is mentioned multiple times but lacks a detailed explanation. The manuscript does not sufficiently elucidate how this effect specifically contributes to the catalyst's performance or structural stability.

[Response]:The term of “high-entropy effect” is discussed in the work by Yeh et al.¹³, where the concept of “high-entropy alloy (HEAs)” is introduced. This concept suggests that the presence of five or more elements in near-equiatomic proportions increases the configurational entropy of mixing by an amount sufficient to overcome the enthalpies of compound formation. In 2016, Ming-Hung Tsai¹⁴ extended this concept to the novel high-entropy intermetallics (HEIs) exhibiting a degree of long-range chemical ordering. Consequently, the physical nature of the “high-entropy effect” refers to the high configurational entropy introduced by multiple element components. (In a solid, entropy arises from contributions from electrons, phonons, and the configuration of mixing. In the field of high-entropy materials, the term “high-entropy” specifically refers to the high configurational entropy due to the mixing of multiple elements.)

Minimizing the Gibbs free energy, ΔG , is the standard method to predict the thermodynamically stable phase in a system of multiple metallic elements¹⁵⁻¹⁶. In such systems, many competing phases exist, and the phase or combination of phases with the lowest Gibbs free energy is the equilibrium state. The mixing Gibbs free energy of our multiple metallic high-entropy alloy (HEA) can be defined as

$$\Delta G_{mix} = \Delta H_{mix} - T\Delta S_{mix} \quad (1)$$

where ΔH_{mix} and ΔS_{mix} are the enthalpy and entropy of mixing, respectively, and T is the absolute temperature. At thermodynamic equilibrium, the phase present in this multiple metallic element system can be stabilized by minimizing the Gibbs free energy of mixing.

In HEI systems, the mixing enthalpy ΔH_{mix} is defined as the difference between the energy of HEI, E_{HEI} , and the ground-state energies of the metal element E_i

$$\Delta H_{mix} = E_{HEI} - \sum_i^n x_i E_i$$

where x_i is the atomic fraction of the element i , and n is the number of metal elements in the system. In our 5-metallic HEI/N-HEI, the mixing enthalpy is calculated to be $\Delta H_{mix} = -0.328$ kJ/mol. Typically, in HEIs, all constituent metal atoms are arranged loosely, resulting in a relatively small ΔH_{mix} . Therefore, the relative stability of HEIs strongly depends on mixing entropy ΔS_{mix} . The ideal entropy of mixing is given by:

$$\Delta S_{mix} = -R \sum_i^n x_i \ln x_i \quad (2)$$

where R is the gas constant. For our 5-metallic HEIs, the ideal mixing entropy is calculated to be $\Delta S_{mix} = 1.386R$. At room temperature ($T=300K$), the contribution of

the mixing entropy to the free energy is $-T\Delta S_{mix} = -3.458$ kJ/mol, which is significantly larger than ΔH_{mix} . Note that single-phase HEIs exhibit long-range chemical ordering and may have a severely distorted lattice, which leads to less mixing entropy than their solid-solution counterparts. Nonetheless, compared to conventional binary or ternary intermetallics, HEIs are not stoichiometric and thus have a configurational mixing entropy much higher than conventional intermetallics with fixed stoichiometry. This high entropic contribution to the free energy at high temperatures T is favorable for the formation of single-phase HEI/N-HEI.

We have added simple and clear definitions and/or explanations for “high-entropy effect” in the Introduction section in the revised Manuscript. (Page 3)

References:

13. Yeh, J. W.; Chen, S. K.; Lin, S. J.; Gan, J. Y.; Chin, T. S.; Shun, T. T.; Tsau, C. H.; Chang, S. Y., Nanostructured High-Entropy Alloys with Multiple Principal Elements: Novel Alloy Design Concepts and Outcomes. *Advanced Engineering Materials* **2004**, *6* (5), 299-303.
14. Tsai, M.-H., Three Strategies for the Design of Advanced High-Entropy Alloys. *Entropy* **2016**, *18* (7).
15. George, E. P.; Raabe, D.; Ritchie, R. O., High-entropy alloys. *Nature Reviews Materials* **2019**, *4* (8), 515-534.
16. Miracle, D. B.; Senkov, O. N., A critical review of high entropy alloys and related concepts. *Acta Materialia* **2017**, *122*, 448-511.

2. Does the high-entropy effect play a primary role in corrosion resistance?

[Response]: In N-HEI/KB, the synergistic interplay of the high-entropy effect, sub-angstrom strain, and the ‘pinning effect’ of M-N bonds contributes to the remarkable stability of N-HEI/KB catalysts. So, it is difficult to determine which one is the primary role in corrosion resistance. A comprehensive analysis of the synergistic interplay of these factors is outlined below:

1. **High-entropy effect:** The presence of “high-entropy alloy” in N-HEI/KB can be proved by the response in above comment 1, which contributes to the catalyst’s thermodynamic stability and can improve overall stability of the catalyst (*Nat. Rev. Mater.* 2019, *4*, 515-534).
2. **Pinning effect:** EXAFS fitting results (Figure R9) confirm the formation of M-N bonds in N-HEI/KB, which contribute to its enhanced stability. Notably, N-HEI/KB

exhibits a significantly lower MA loss (17.9%) compared to HEI/KB (26.4%) after 30k ADT cycles (Figure R2), supporting that the ‘pinning effect’ of M-N bonds plays a crucial role in corrosion resistance.

3. **Sub-angstrom strain:** There is more displacement of TMs in the N-HEI/KB than in PtNiN/KB (Reviewer 2, Comment 2), confirming the unique sub-angstrom strain in the N-HEI structure. The higher ORR stability observed for N-HEI/KB compared to PtNiN/KB (Figure R2) further suggests that this sub-angstrom strain contributes significantly to corrosion resistance.

While current data and characterizations support the involvement of all above three factors, it is challenging to determine which one is the primary contributor to corrosion resistance. The synergistic interplay of these factors likely underlies the enhanced stability of N-HEI/KB catalysts.

Figure R13. DFT and uMLIPs calculations of the N-HEI nanoparticle. (a) Cohesive energy of the L10 unit cell calculated by fine-tuned universal Machine Learning Interatomic Potentials (uMLIPs). (b) The average relative probability (RPP) parameters for Co, Fe, Ni, and Cu elements up to the 3rd NN shell. RPP = 1 indicates that the two species are as likely to be found within a given neighboring shell as any atom pairs. RPP > 1 (< 1) indicates that they are more (less) likely to be found in that neighboring shell. (c) Side and top view of the L10-HEI model (Pt: gray, Co: pink, Ni: green, Fe:

orange; Cu: brown). (d) Pt vacancy diffusion energy of L1₀-HEI with/without N-doping and L1₀-PtNi using uMLIPs methods. (e) The diffusion barrier path and activation energy of Pt vacancy for L1₀-HEI with/without N-doping using DFT calculations. (f) Schematic diagram of Pt-vacancy-mediated diffusion of TM atoms in the L1₀-HEI with N slab model.

3. Are there variations in the contributions of the "pinning effect" and sub-angstrom strain under different cycling conditions?

[Response]: Yes, both the “pinning effect” and sub-angstrom strain are expected to change under different cycling conditions. However, based on the current data and characterization techniques, we can only offer qualitative conclusions rather than a detailed quantitative analysis.

From above response for comment 2, it is clear that both the ‘pinning effect’ of M-N bonds and sub-angstrom strain contribute to the ORR performance of N-HEI/KB catalysts. After MEA testing, the elemental mapping of a representative N-HEI nanoparticle reveals that N elements are maintained within the core structure (Figure R14), suggesting the presence of M-N bonds. However, due to the dissolution of TMs during the ORR process, the number of M-N bonds decreases (Table R2), leading to a reduction in the ‘pinning effect’ as cycling progresses.

The atomic displacement of TMs is still observed in the N-HEI NP after ADT cycles (Figure R15), indicating the presence of sub-angstrom strain in N-HEI during the MEA testing. The average FWHM ratio of TM columns to Pt columns is 0.95 ± 0.03 , which is slightly lower than that of BOL value of 0.98, suggesting that the sub-angstrom strain may slightly decrease at EOL.

Based on above analysis, we can conclude that the contributions of the “pinning effect” and sub-angstrom strain decrease slightly with increasing cycling but both are still effective.

Figure R14. Structure and morphology of N-HEI/KB catalyst after ADT cycles in MEA testing. a. STEM-HAADF image and b. the particle size distribution of the N-HEI/KB catalyst after AST 90k cycles. c. STEM-HAADF image and the EDS elemental mappings of the N-HEI/KB nanoparticles after ADT 90k cycles.

Figure R15. a. The same atomic-resolution STEM-HAADF image as Fig. 3c (N-HEI nanoparticle after ADT 90k cycles in MEA testing) with TM displacement arrow map embedded. b. Magnified image from the yellow rectangle in (a). c-d. Additional STEM-HAADF images from N-HEI nanoparticle after ADT 90k cycles in MEA testing, showing multiple layers Pt shell. Scale bar 2 nm for (a,c) and 0.5 nm for (b,d).

4. For the quantification of sub-angstrom strain, can additional experimental or theoretical data be provided to further validate the findings?

[Response]: Thank you for your comment. We have calculated the site-dependent local atomic distortion in our L1₀-HEI and L1₀-N-HEI models, and compared these results with those for L1₀-PtNi intermetallics, as shown in Figure R16.

The site-dependent local atomic distortion is defined as the degree of atomic displacement from the ideal lattice sites $\Delta\mathbf{r}_i = \mathbf{r}'_i - \mathbf{r}_i$, where \mathbf{r}_i and \mathbf{r}'_i are the atomic coordinates of site i before and after lattice distortion, respectively. Note that the structural distortion of crystalline materials typically involves both lattice structure distortion (variations in the lattice structure) and the local atomic distortion (atomic displacements from ideal lattice sites). In our calculation, $\Delta\mathbf{r}_i$ accounts only for local atomic distortion, as this is the primary contributor to sub-angstrom strain in this system. The greater the divergence in local atomic distortion around neighboring atoms, the stronger the sub-angstrom strain observed in experiments. In Figure R16, we show the mapping plot of site-dependent local atomic distortion in the Pt and TM layers of HEI, N-HEI and PtNi intermetallics. With the nearly random occupancy of different transition metals, the local atomic distortion in the TM layer in the HEI ($\Delta r_{max} = 0.037 \text{ \AA}$) is greatly larger than that in the PtNi binary intermetallics ($\Delta r_{max} = 0.002 \text{ \AA}$). After N-doping, even more severe atomic distortion is observed in the N-HEI system ($\Delta r_{max} = 0.092 \text{ \AA}$). Similar trend is observed for Pt layer, but the magnitude of local atomic displacement of Pt layer is smaller than TM layer. This severe atomic distortion increases the energy barrier for atoms to be activated and migrated, thus reducing the diffusion rate of atoms.

These results have been added to the revised Supplementary Information (Supplementary Fig. 25). The corresponding description was added to the revised Manuscript (Page 15).

Figure R16. The mapping plot of site-dependent local atomic distortions in L1₀-HEI, L1₀-N-HEI and L1₀-PtNi intermetallics. The local lattice distortions in the TM layer are projected onto the (111) plane for (a) L1₀-HEI, (b) L1₀-N-HEI, and (c) L1₀-PtNi, and those in the Pt layer for (d) L1₀-HEI, (e) L1₀-N-HEI, and (f) L1₀-PtNi. The magnitude and direction of the local atomic distortion are indicated by arrows, and the max local atomic displacement are shown for each structure. Note that we used fractional coordinates and magnified the atomic displacements with the same scale for all plots to improve visualization. (Pt: gray, Co: pink, Ni: green, Fe: orange, Cu: brown, N: blue)

5. Regarding the stability of the catalyst under MEA testing, a comparative analysis with conventional catalysts over extended cycles would enhance the credibility of the results.

[Response]: Thank you for your suggestion. We added conventional catalysts over extended cycles and other HEI/HEA catalysts for comparison in revised Supplementary Information (Supplementary Table 8). As shown in Table R6, the MEA performance of N-HEI/KB outperformed most of the reported Pt-based intermetallic catalysts reported in recent literature.

Table R6. Summary of MEA performance of the recently reported Pt-based electrocatalysts. *Data extracted from plots in the literatures.

	Cathode loading (mg _{Pt} cm ⁻²)	Current density @ 0.7 V (A cm ⁻²)		Voltage loss @0.8 A cm ⁻² (mV)	Test Conditions (temperature, pressure)	Reference
		BOL	EOL			
N-HEI/KB	0.2	1.52	1.39 (90k)	9 (90k)	80°C, 250 kPa _{abs}	This work
Pt (40wt%)/Mn-N-C	0.25	1.41	1.20 (150k)	----	80°C, 250 kPa _{abs}	Ref ¹⁷
L1 ₂ -Pt ₃ Co@Mn _{SA} -NC	0.2	1.75	1.43 (90k)	30* (90k)	80°C, 250 kPa _{abs}	Ref ¹⁸
PtCo/KB-NH ₂	0.2	1.31	0.83 (90k)	30 (30k)	80°C, 150 kPa _{abs}	Ref ¹⁹
Gd-O-Pt ₃ Ni	0.1	1.24	0.81 (50k)	19 (40k)	80°C, 200 kPa _{abs}	Ref ¹¹
i-CoPt/KB	0.1	1.5*	1.25* (30k)	29 (30k)	80°C, 250 kPa _{abs}	Ref ²⁰
PtCo/Gnp	0.06	1.22*	1.06* (30k)	23.5 (30k)	80°C, 150 kPa _{abs}	Ref ²¹
Pt ₃ Co/FeN ₄ -C	0.1	1.05*	0.8* (30k)	22 (30k)	80°C, 150 kPa _{abs}	Ref ²²
STG-PtCo	0.025	1.2*	1.0* (30k)	21 (30k)	80°C, 150 kPa _{abs}	Ref ²³
L1 ₀ -PtZn	0.1	0.96	0.91 (30k)	10 (30k)	80°C, 150 kPa _{abs}	Ref ²⁴
PtNi/C3	0.1	1.0*	0.7* (30k)	16 (30k)	94°C, 250 kPa _{abs}	Ref ²⁵
40%L1 ₀ -Pt ₅₀ Ni ₃₅ Ga ₁₅ /C	0.2	1.67	1.33 (90k)	25 (90k)	80°C, 250 kPa _{abs}	Ref ²⁶
PtNiMoAu NWs/C	0.1	0.93	0.72 (30k)	25 (30k)	80°C, 150 kPa _{abs}	Ref ²⁷
N-Pt/HEA/C	0.2	1.64	1.44 (30k)	18 (30k)	80°C, 250 kPa _{abs}	Ref ⁷

References:

- 7 Zhao, X. *et al.* Multiple Metal-nitrogen bonds synergistically boosting the activity and durability of high-entropy alloy electrocatalysts. *J. Am. Chem. Soc.* **146**, 3010-3022 (2024).
- 11 Yang, L. *et al.* Rare earth evoked subsurface oxygen species in platinum alloy catalysts enable durable fuel cells. *Angew. Chem. Int. Ed.* **63**, e202315119 (2024).
- 17 Zeng, Y. *et al.* Pt nanoparticles on atomic-metal-rich carbon for heavy-duty fuel cell

- catalysts: durability enhancement and degradation behavior in membrane electrode assemblies. *ACS Catal.* **13**, 11871-11882 (2023).
- 18 Zeng, Y. *et al.* Regulating catalytic properties and thermal stability of Pt and PtCo intermetallic fuel-cell catalysts via strong coupling effects between single-metal site-rich carbon and Pt. *J. Am. Chem. Soc.* **145**, 17643-17655 (2023).
- 19 Gong, Q. *et al.* Amino-tethering synthesis strategy toward highly accessible sub-3-nm L1₀-PtM catalysts for high-power fuel cells. *Matter* **6**, 963-982 (2023).
- 20 Yoo, T. Y. *et al.* Scalable production of an intermetallic Pt–Co electrocatalyst for high-power proton-exchange-membrane fuel cells. *Energy Environ. Sci.* **16**, 1146-1154 (2023).
- 21 Zhao, Z. *et al.* Graphene-nanopocket-encaged PtCo nanocatalysts for highly durable fuel cell operation under demanding ultralow-Pt-loading conditions. *Nat. Nanotechnol.* **17**, 968-975 (2022).
- 22 Qiao, Z. *et al.* Atomically dispersed single iron sites for promoting Pt and Pt₃Co fuel cell catalysts: performance and durability improvements. *Energy Environ. Sci.* **14**, 4948-4960 (2021).
- 23 Song, T. W. *et al.* Small molecule-assisted synthesis of carbon supported platinum intermetallic fuel cell catalysts. *Nat. Commun.* **13**, 6521 (2022).
- 24 Liang, J. *et al.* Biaxial strains mediated oxygen reduction electrocatalysis on fenton reaction resistant L1₀-PtZn fuel cell cathode. *Adv. Energy Mater.* **10**, 2000179 (2020).
- 25 Zhao, Z. *et al.* Tailoring a three-phase microenvironment for high-performance oxygen reduction reaction in proton exchange membrane fuel cells. *Matter* **3**, 1774-1790 (2020).
- 26 Liang, J. *et al.* Metal bond strength regulation enables large-scale synthesis of intermetallic nanocrystals for practical fuel cells. *Nat. Mater.* **23**, 1259-1267 (2024).
- 27 Gao, L. *et al.* Identifying the distinct roles of dual dopants in stabilizing the platinum-nickel nanowire catalyst for durable fuel cell. *Nat. Commun.* **15**, 508 (2024).

Response to the reviewers' comments

Dear referees,

We sincerely appreciate the reviewers for making constructive comments and valued suggestions, which greatly help us to improve the quality of our manuscript. In the following, we list the point-by-point response to the reviewers' comments. In this Response Letter, the replies are in blue text and the figures are named as Figure R*. All changes have been highlighted by yellow color in the revised Manuscript and Supplementary Information.

REVIEWER COMMENTS

Reviewer #1 (Remarks to the Author):

While the authors have responded to many of the concerns raised in the 1st review, some points still remain open:

1. The authors have included results obtained from Rietveld refinements of the HEI formation, but not of the N-HEI sample. Unfortunately, the Rietveld refinements are not shown, also the refined parameters and a goodness of fit is not reported. This must be included as without these data it is not possible to judge whether the model actually fit the data. And it is unclear why a two-phase refinement was necessary from 200-400 degrees C.

[Response]: We sincerely appreciate the reviewer's valuable comments. We have added the Rietveld refinements of the N-HEI sample (Figure R1h), refined parameters, and goodness of fit (R_{wp}).

In this study, Rietveld refinement was performed on the *in situ* XRD patterns collected at different thermal treatment stages to analyze the phase evolution using TOPAS software (*Journal of Applied Crystallography*, 51, 210-218, 2018). For stage I (<220 °C), the XRD patterns were refined using a cubic Pt phase with the *Fm-3m* space group. In stage II (220–740 °C), the patterns were fitted with a combination of cubic Pt (*Fm-3m*) and HEA (*Fm-3m*) phases. At stage III (>740 °C), the refinement was conducted using both the cubic HEA phase (*Fm-3m*) and a tetragonal HEI phase with the *P4/mmm* space group, indicative of L1₀-type ordering. During high-temperature holding period (850°C), the refinement was conducted using the tetragonal HEI phase

($P4/mmm$). To simplify the refinement, Co, Ni, Fe, and Cu were grouped as a single transition metal component, and PtNi was used to represent the HEA/HEI structures for Rietveld refinements in TOPAS. Representative Rietveld refinement results from each stage were selected for illustration, as shown in Figure R1b–d.

The weighted profile R-factor (R_{wp}) values for the *in-situ* Rietveld refinements ranged from 17% to 25%, indicating an acceptable level of agreement between the observed and calculated XRD profiles. The R_{wp} value for the HEI/KB sample annealed at 850 °C for 60 min is as low as 2.4% (Figure R1f), indicating an excellent quality of fit. Additionally, Rietveld refinement of the N-HEI sample was performed using a two-phase model comprising the cubic Pt phase and the tetragonal HEI phase, yielding an R_{wp} of 4.8% (Figure R1h), further confirming the high reliability of the fitting results.

Figure R1 has been added to the revised Supplementary Information as new Supplementary Fig.1. The corresponding description can be found on pages 4-5 in the revised Manuscript and pages 2-3 in revised Supplementary Information.

And it is unclear why a two-phase refinement was necessary from m 200-400 degrees.

As we discussed above, there are two phases coexisting at both stage II (Pt and HEA phases) and stage III (HEA and HEI phases). However, it is worth noting that the diffraction peaks corresponding to the Pt phase became significantly weaker above 400 °C, resulting in increased uncertainty in the refined lattice parameters. Therefore, only the lattice constants of Pt obtained below 400 °C are presented in Supplementary Fig. 1a. We have added this description to the revised Supplementary Information (Page 3).

Figure R1. a. Rietveld refinement analysis showing changes in lattice constant of the HEI nanoparticles as a function of annealing temperatures. b-d. Rietveld refinement of XRD patterns of HEI/KB at different annealing temperature: b. 200 °C, c. 600 °C, and d. 750 °C. e. *In situ* synchrotron XRD patterns of HEI/KB holding at 850 °C in 10% H₂/He gas stream (wavelength 0.6199 Å). f. Rietveld refinement of XRD patterns of HEI/KB holding at 850 °C for 60 min. g. XRD patterns of N-HEI/KB and HEI/KB catalysts (Cu K α , wavelength 1.5406 Å). h-i. Rietveld refinement of XRD patterns of N-HEI/KB and HEI/KB in (g). The “Diff” curve represents the difference between the experimentally measured and the calculated XRD patterns, highlighting the quality of the Rietveld refinement fit.

2. The diffraction pattern in Figure 8c do not match the ones from the in-situ experiments. In Figure 8c a clear shoulder is visible which indicates a phase separation, which is most pronounced for the N-HEI samples. Can the authors comment on that?

[Response]: We thank the reviewer’s insightful comment. We’d like to clarify that the *in situ* XRD measurements were primarily employed to track the phase evolution during thermal treatment. Under prolonged annealing at sufficiently high temperatures, the

particles are expected to fully convert into the HEI phase. However, such extended treatment often leads to significant particle growth (for example, the particle size of HEI/KB at 850 °C for 60 min is about 32 nm), which impacts the ORR performance due to reduced surface area. In order to maximize the utilization of catalysts, we optimized the annealing condition to control the particle size around 3 nm. Under these optimized conditions, the dominant phase is expected to be HEI with a monolayer Pt shell. Nonetheless, the shoulder of N-HEI/KB and HEI/KB in Figure R1g can be reasonably attributed to the small HEI particles with Pt-rich shells.

To validate this hypothesis, we performed Rietveld refinement on both the N-HEI/KB and HEI/KB samples using a two-phase model comprising the cubic Pt phase (*Fm-3m*) and tetragonal HEI phase (*P4/mmm*). As shown in Figure R1h–i, the excellent agreement between observed and calculated patterns, as evidenced by the low R_{wp} values (4.8% for N-HEI/KB and 4.6% for HEI/KB), supports our interpretation of a mixed-phase structure under size-controlled annealing conditions. The relative description has been added in the revised Manuscript (Page 5).

3. Also better quality diffraction data as well as Rietveld refinement of these samples (which I assume are the ones actually used for the catalysis) are still missing. Here still the "strain" is shown by drawing lines into a very broad and unsymmetric diffraction peak. This is not sufficient.

[Response]: We sincerely appreciate the reviewer's valuable comments. In response to your suggestion, we have provided higher-quality diffraction data and performed Rietveld refinement for the N-HEI/KB and HEI/KB samples. The updated results are presented in above Figure R1g–i.

The XRD results indicate a shift in the peak positions of N-HEI/KB towards lower angles compared to those of HEI/KB (Figure R1g), this suggests successful doping of N atoms with resultant tensile strain in HEI. Furthermore, the Rietveld refinement results (Figure R1h–i and Table R1) showed the refined lattice parameters of $3.680 \pm 0.003 \text{ \AA}$ for HEI phase in N-HEI/KB, which is larger than that of HEI/KB ($3.675 \pm 0.003 \text{ \AA}$ for HEI phase). These results confirm the strain modulation induced by N doping.

These results have been added to the revised Supplementary Fig.1 and Table 3. The corresponding description can be found in the revised Manuscript (Page 5).

Table R1. Parameters and R-values obtained from Rietveld refinements for the samples.

Samples	phase	Ratio (wt.%)	Lattice parameter		R _{wp}
			a=b	c	
N-HEI/KB	HEI	54.1	3.680 ± 0.003 Å	2.710 ± 0.002 Å	4.8%
	Pt	45.9	3.902 ± 0.005 Å	3.902 ± 0.005 Å	
HEI/KB	HEI	59.4	3.675 ± 0.003 Å	2.710 ± 0.002 Å	4.6%
	Pt	40.6	3.899 ± 0.006 Å	3.899 ± 0.006 Å	

4. The EXAFS fitting shows that the Pt-TM distance is virtually the same, while the Pt-Pt distance varies for N-HEI and HEI. However, the HAADF analysis shows that the Pt-TM distance varies. Can the authors comment on this discrepancy. What is the uncertainty of the HAADF TM position fitting?

[Response]: We thank the reviewer's insightful comment. The EXAFS fitting gives the average Pt-TM and Pt-Pt distance over many particles and all directions, while atomic resolution HAADF analysis shows displacement (amplitude and direction) of the individual TM column in the individual particle. We do observe the Pt-TM distance varies due to the displacement of TM from the lattice center. However, the direction of the displacement of the TMs in the particle is quite random, thus the averaged Pt-TM distance over many particles would be invariant.

The uncertainty of the HAADF TM position fitting is approximately 0.02 Å. We have added this note to the caption of Fig. 1 in the revised Manuscript. (Page 8)

5. In the ECSA determination, CO stripping should be performed and compared to the Hupd, as the determination of surface area is challenging for the multi-component alloys and should not rely on a single method.

[Response]: We appreciate the reviewer's valuable comment. Following the reviewer's suggestion, we have added the CO stripping curves of N-HEI/KB, HEI/KB and commercial Pt/C. CO stripping test was conducted by bubbling CO gas into 0.1 M HClO₄ electrolyte for 30 min while maintaining the electrode potential at 0.05 V. Then Ar gas was purged into the electrolyte for another 30 min to remove unadsorbed CO in the electrolyte. CV curve was then collected by scanning from 0.05 V to 1.05 V at a scanning rate of 50 mV s⁻¹. The ECSA was calculated accordingly:

$$ECSA = \frac{S_{CO}}{0.42 \times V_{scan} \times m_{Pt}}$$

where S_{CO} , V_{scan} , and m_{Pt} represent the integral area of CO stripping ($\text{mA}\cdot\text{V}$), scanning rate (mV s^{-1}), and Pt loading (μg), respectively. We also used an area-specific charge of two-electron transfer, 0.42 mC cm^{-2} .

Based on the CO stripping measurements, the ECSA were quantified to be $77.6 \text{ m}^2 \text{ g}_{Pt}^{-1}$, $69.4 \text{ m}^2 \text{ g}_{Pt}^{-1}$, and $85.7 \text{ m}^2 \text{ g}_{Pt}^{-1}$ for N-HEI/KB, HEI/KB, and commercial Pt/C, respectively (Figure R2). The trend observed in ECSAs obtained via CO stripping is consistent with that determined by the H_{upd} method ($61.2 \text{ m}^2 \text{ g}_{Pt}^{-1}$ for N-HEI/KB, $54.8 \text{ m}^2 \text{ g}_{Pt}^{-1}$ for N-HEI/KB, and $83.3 \text{ m}^2 \text{ g}_{Pt}^{-1}$ for commercial Pt/C). The ECSAs of N-HEI/KB and HEI/KB determined by the CO stripping are 1.27 times higher than those by the H_{upd} method, possibly because of a suppression effect of sublayer transition metals on H_{upd} (Angew. Chem., Int. Ed. 51, 3139–3142, 2012; Electrocatalysis, 5, 408–418, 2014). It is interesting to investigate the origin of the difference in ECSA between the H_{upd} and CO stripping methods in multi-elemental Pt-based materials, but we consider that it would be a separate study in electrocatalysis through experimental and theoretical approaches in the future.

These results have been added in the revised Supplementary Information (Supplementary Fig. 7). The corresponding description can be found on page 9 and pages 19-20 in the revised Manuscript.

Figure R2. ORR performance of N-HEI/KB, HEI/KB, and commercial Pt/C in RDE testing. a. CV curves of N-HEA/C at beginning-of-life (BOL) and different ADT cycles. b. MA and ECSA of N-HEI/KB catalyst at BOL and different potential cycles. c. CO stripping curve of N-HEI/KB. d-e. ORR polarization curves and CV curves of HEI/KB at BOL and ADT 30,000 cycles. f. CO stripping curve of HEI/KB. g-h. ORR polarization curves and CV curves of commercial Pt/C at BOL and ADT 30,000 cycles. i. CO stripping curve of commercial Pt/C.

6. While the authors have added ICP-MS data which were requested, the manuscript still relies on STEM-EDS for quantification of the elements. STEM-EDS is a technique which is not well suited for this purpose and needs to be done with utmost care, see e.g. <https://www.sciencedirect.com/science/article/pii/S0968432818301021#bib0190>

[Response]: We sincerely thank the reviewer for the insightful comment. We fully agreed that STEM-EDS has inherent limitations in terms of quantification accuracy. The technical details of the STEM-EDS quantification have been included in our response to the following comment. We would also like to highlight that we have incorporated ICP-MS data in a previously revised manuscript, which provides precise bulk elemental compositions and serves to validate the results obtained by STEM-EDS. As shown in Supplementary Tables 1-2, the elemental ratios obtained from STEM-EDS show very good agreement with those measured by ICP-MS, indicating that the STEM-

EDS data reported in this study are reasonable and reliable within the context of local compositional analysis. Importantly, the conclusions regarding elemental composition (atomic ratio in N-HEI/KB and HEI/KB: Pt:Co:Ni:Fe:Cu=4:1:1:1:1) are drawn with reference to the quantitative ICP-MS results, not solely based on STEM-EDS.

In this study, the STEM-EDS technique was used to examine the elemental distribution and local compositional uniformity at the nanoscale (Supplementary Figs. 2-3). And it was also used to compare relative compositional changes before and after RDE stability testing for all catalysts (Supplementary Figs. 9-10 and Table 7). The catalyst amount (including carbon) on a RDE is approximately 10~12 μg , which is clearly insufficient for ICP-MS analysis that generally requires a few ten mg. Therefore, we believe that the STEM-EDS technique is suitable for these two purposes.

7. Please comment on how quantitative information were obtained from the microscopy and add information on the errors of this estimation.

[Response]: Thank you for your insightful comment. We have included a description of how quantitative information was obtained from the microscopy data. In this study, we utilized a FEI Talos F200X microscope equipped with a four-quadrant, 0.9-sr solid-angle EDS detector, enabling fast and high-precision elemental mapping. Specifically, chemical compositions were extracted from STEM-EDS mapping using integrated X-ray intensities corrected by the Cliff–Lorimer method with standardless quantification. To ensure accuracy, we applied absorption and background corrections using the software's built-in ZAF algorithm. We selected representative catalyst particles and conducted multiple measurements under identical conditions. The probe current and dwell time were optimized to balance spatial resolution and signal-to-noise ratio, while minimizing beam-induced damage.

To estimate the errors, we measured multiple regions (≥ 10 regions per sample) and reported the average elemental compositions. The relative error of the STEM-EDS quantification was typically within $\pm 2\text{--}4$ at.% for the main metallic elements (e.g., Pt), showing good agreement with the ICP-MS data (Supplementary Tables 1-2), which confirms the reliability of the microscopy-derived values within their error margins. Nevertheless, we emphasize that our compositional conclusions are based on the combination of STEM-EDS and ICP-MS data. The STEM-EDS served for analyzing local compositional uniformity, spatial distribution, and relative changes before and after durability testing.

The corresponding description has been added in the revised Manuscript (Page 19) and Supplementary Information (Page 28).

Reviewer #2 (Remarks to the Author):

The author has effectively addressed my concerns regarding the manuscript.

[Response]: Many thanks to the reviewer for recognizing this work!

Reviewer #3 (Remarks to the Author):

The authors have made commendable efforts to address most of the concerns raised in the previous review, and their responses have largely resolved most doubts.

However, I believe the concept of high-entropy and its core effects could still be further elaborated and introduced in greater detail. Given the widespread application of high-entropy strategies across various materials, a more in-depth discussion would enhance the manuscript's scientific rigor and contextual depth. For instance, the authors may consider incorporating insights from studies such as *Energy & Environmental Science* (2021, 14(5): 2883-2905) and *Energy & Environmental Science* (2025, 18: 19-52). Additionally, regarding the calculation of high-entropy sites, it appears that the possibility of multiple-site configurations has been overlooked. Further elaboration and clarification on this aspect would strengthen the discussion.

[Response]: We sincerely thank the reviewer for the thoughtful and constructive feedback, as well as for acknowledging our efforts to address the previous concerns. We fully agree that a more detailed discussion on the fundamental concept of high-entropy and its implications across various materials would enhance the scientific depth of the manuscript. We also appreciate the reviewer's recommendation to include insights from relevant literature. In the revised Manuscript, we have expanded the Introduction section to provide a more comprehensive overview of high-entropy and its core effects. (Page 3)

Additionally, regarding the calculation of high-entropy sites, it appears that the possibility of multiple-site configurations has been overlooked. Further elaboration and clarification on this aspect would strengthen the discussion.

We thank the reviewer for pointing out the need to clarify the treatment of multiple-site

configurations in our HEIs calculations. To effectively explore the vast configurational space of HEIs, we employed a two-step approach that integrates universal Machine Learning Interatomic Potentials (uMLIPs) with density functional theory (DFT) calculations, aiming to balance computational efficiency and accuracy. To develop accurate uMLIPs, we constructed a comprehensive dataset using DFT simulations of structures ranging from pure and binary to quinary compositions, with and without nitrogen interstitials, and across various supercell sizes. These structures were designed to capture a wide range of possible multi-site configurations in the HEI system, as detailed in the Methods section and Supplementary Fig. 20. The fine-tuned uMLIPs demonstrate high predictive performance across key metrics including energy, force, stress, lattice parameters, and volume (Supplementary Tables 11-12). Using the trained uMLIPs, we efficiently screened 500 randomly generated HEI configurations and identified the L1₀-HEI structure with the lowest potential energy. This structure was then used for further property evaluations, including the calculation of Pt vacancy formation energies and diffusion barriers with and without nitrogen interstitials. To verify the nitrogen-induced sluggish diffusion mechanism, we also performed DFT-based climbing image nudged elastic band (CI-NEB) calculations to evaluate Pt vacancy migration barriers.

In response to the reviewer's comment, we have revised the Methods section to explicitly describe our treatment of site-specific compositional disorder in HEIs (Pages 23-24). And the relative discussion has been added to the revised Manuscript (Pages 14). These additions aim to strengthen the theoretical and methodological underpinnings of our study, and we hope they address the reviewer's concerns satisfactorily.